# Non-Adversarial Inverse Reinforcement Learning via Successor Feature Matching

**Arnav Kumar Jain**[1,2,*] **Harley Wiltzer**[1,3,] 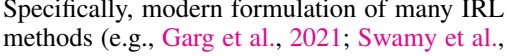**Jesse Farebrother**[1,3,] 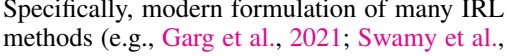
**Irina Rish**[1,2] **Glen Berseth**[1,2] **Sanjiban Choudhury**[4]
[1]Mila – Québec AI Institute  [2]Université de Montréal  [3]McGill University  [4]Cornell University

## ABSTRACT

In inverse reinforcement learning (IRL), an agent seeks to replicate expert demonstrations through interactions with the environment. Traditionally, IRL is treated as an adversarial game, where an adversary searches over reward models, and a learner optimizes the reward through repeated RL procedures. This game-solving approach is both computationally expensive and difficult to stabilize. In this work, we propose a novel approach to IRL by *direct policy search*: by exploiting a linear factorization of the return as the inner product of successor features and a reward vector, we design an IRL algorithm by policy gradient descent on the gap between the learner and expert features. Our non-adversarial method does not require learning an explicit reward function and can be solved seamlessly with existing RL algorithms. Remarkably, our approach works in state-only settings without expert action labels, a setting which behavior cloning (BC) cannot solve. Empirical results demonstrate that our method learns from as few as a single expert demonstration and achieves improved performance on various control tasks.

## 1 INTRODUCTION

In imitation learning (Abbeel & Ng, 2004; Ziebart et al., 2008; Silver et al., 2016; Ho & Ermon, 2016; Swamy et al., 2021), the goal is to learn a decision-making policy that reproduces *behavior* from demonstrations. Rather than simply mimicking the state-conditioned action distribution as in behavior cloning (Pomerleau, 1988), interactive approaches like Inverse Reinforcement Learning (IRL; Abbeel & Ng, 2004; Ziebart et al., 2008) have the more ambitious goal of synthesizing a policy whose long-term occupancy measure approximates that of the expert demonstrator by some metric. As a result, IRL methods have proven to be more robust, particularly in a regime with few expert demonstrations, and has lead to successful deployments in real-world domains such as autonomous driving (e.g., Bronstein et al., 2022; Vinitsky et al., 2022; Igl et al., 2022). However, this robustness comes at a cost: IRL approaches tend to involve a costly bi-level optimization.

Specifically, modern formulation of many IRL methods (e.g., Garg et al., 2021; Swamy et al.,

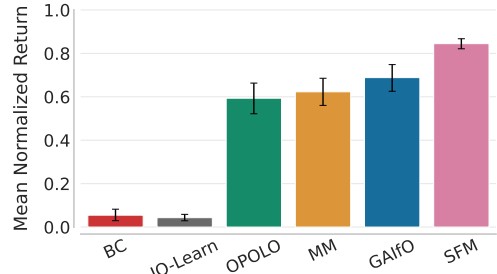

Figure 1: Comparing Mean Normalized Return on 10 tasks from DeepMind Control suite (Tunyasuvunakool et al., 2020) of our method SFM against offline Behavior Cloning (Pomerleau, 1988), the non-adversarial IRL method IQ-Learn (Garg et al., 2021), and the state-only adversarial methods OPOLO (Zhu et al., 2020), MM (Swamy et al., 2021) and GAIfO (Torabi et al., 2018), where the agents are provided a single expert demonstration. Our state-only non-adversarial method SFM achieves a higher mean normalized return. Error bars show the 95% bootstrap CIs.

---

*Correspondence to `arnav-kumar.jain@mila.quebec`.

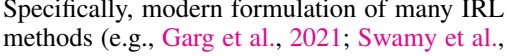Author order decided by chili-eating contest; result was inconclusive.

Our codebase is available at https://github.com/arnavkj1995/SFM.

2021) involve a min-max game between an adversary that learns a reward function to maximally differentiate between the agent and expert in the outer loop and a Reinforcement learning (RL) subroutine over this adversarial reward in the inner loop. However, all such methods encounter a set of well-documented challenges: (1) optimizing an adversarial game between the agent and the expert can be unstable, often requiring multiple tricks to stabilize training (Swamy et al., 2022), (2) the inner loop of this bi-level optimization involves repeatedly solving a computationally expensive RL problem (Swamy et al., 2023), and (3) the reward function class must be specified in advance. Moreover, many approaches to imitation learning require knowledge of the actions taken by the demonstrator. This renders many forms of demonstrations unusable, such as videos, motion-capture data, and generally any demonstrations leveraging an alternative control interface than the learned policy (e.g., a human puppeteering a robot with external forces). As such, it is desirable to build IRL algorithms where the imitation policies learn from only expert states.

These challenges lead us to the following research question: *Can a non-adversarial approach to occupancy matching recover the expert's behavior without action labels?* To address this question, we revisit the earlier approaches to *feature matching* (Abbeel & Ng, 2004; Ziebart et al., 2008; Syed & Schapire, 2007; Syed et al., 2008), that is, matching the accumulation of discounted state or state-action features along the expert's trajectory. For this task, we propose to estimate expected cumulative sum of features using Successor Features (SF; Barreto et al., 2017) – a low-variance, fully online algorithm that employs temporal-difference learning. Leveraging the benefits of SFs, we demonstrate that *feature matching can be achieved by direct policy search* via policy gradients. In doing so, we present a new approach to IRL, called Successor Feature Matching (SFM), which provides a remarkably simple algorithm for imitation learning.

Interestingly, when the learned features are action-independent, we show that SFM can imitate an expert without knowledge of demonstrators' actions. This accommodates a variety of expert demonstration formats, such as video or motion-capture, where action labels are naturally absent. Additionally, rather than manually pre-specifying a class of expert reward functions (Swamy et al., 2021), SFM *adaptively* learns this class from data using unsupervised RL techniques. Our experiments validate that SFM successfully learns to imitate from as little as a single expert demonstration. As a result, SFM outperforms its competitors by **16%** on mean normalized returns across a wide range of tasks from the DMControl suite (Tunyasuvunakool et al., 2020) —as highlighted in Figure 1. To summarize, the contributions of this work are as follows:

1. **Occupancy matching via reduction to reinforcement learning.** In this work, we propose an algorithm for *feature matching that can be achieved by direct policy search* via policy gradients for inverse RL. In doing so, our method Successor Feature Matching (SFM) achieves strong imitation performance using any off-the-shelf RL algorithms.

2. **Imitation from a single state-only demonstration.** Our method learns with demonstrations without expert action labels by using state-only features to estimate the SFs. To our knowledge, SFM is the *only* online method capable of learning from a single unlabeled demonstration without requiring an expensive and difficult-to-stabilize bilevel optimization (Swamy et al., 2022).

## 2 RELATED WORK

**Inverse Reinforcement Learning (IRL)** methods typically learn via adversarial game dynamics, where prior methods assume base features are known upfront (Abbeel & Ng, 2004; Ziebart et al., 2008; Syed & Schapire, 2007; Syed et al., 2008). The advent of modern deep learning architectures led to methods that do not estimate expected features, but instead learn a more expressive reward function that captures the differences between the expert and the agent (Ho & Ermon, 2016; Swamy et al., 2021; Fu et al., 2018). The class of Moment Matching (MM; Swamy et al., 2021) methods offers a general framework that unifies existing algorithms through the concept of moment matching, or equivalently Integral Probability Metrics (IPM; Sun et al., 2019). In contrast to these methods, our approach is non-adversarial and focuses on directly addressing the problem of matching expected features. Furthermore, unlike prior methods in Apprenticeship Learning (AL; Abbeel & Ng, 2004) and Maximum Entropy IRL (Ziebart et al., 2008), our work *does not* assume the knowledge of base features. Instead, SFM leverages representation learning technique to extract relevant features from the raw observations. The method most similar to ours is IQ-Learn (Garg et al., 2021), a non-adversarial approach that utilizes an inverse Bellman operator to directly estimate the value

function of the expert. Our method is also non-adversarial but offers a significant advantage over IQ-Learn: it does not require knowledge of expert actions during training — providing a state-only imitation learning algorithm (Torabi et al., 2019). However, many existing state-only methods also rely on adversarial approaches (Torabi et al., 2018; Zhu et al., 2020). For instance, GAIfO (Torabi et al., 2018) modifies the discriminator employed in GAIL (Ho & Ermon, 2016) to use state-only inputs, while OPOLO (Zhu et al., 2020) combines a similar discriminator with an inverse dynamics model to predict actions for expert transitions to regularize the agent. Similarly, R2I (Gangwani & Peng, 2020) proposed learning an indirect function to enable imitation when the transition dynamics change. In contrast, SFM is a non-adversarial method that learns from state-only demonstrations.

**Successor Features (SF; Barreto et al., 2017)** generalize the idea of the successor representation (Dayan, 1993) by modeling the expected cumulative state features discounted according to the time of state visitation. Instead of employing successor features for tasks such as transfer learning (Barreto et al., 2017; Lehnert et al., 2017; Barreto et al., 2018; Abdolshah et al., 2021; Wiltzer et al., 2024b;a), representation learning (Le Lan et al., 2022; 2023b;a; Farebrother et al., 2023; Ghosh et al., 2023), exploration (Zhang et al., 2017; Machado et al., 2020; Jain et al., 2023), or zero-shot RL (Borsa et al., 2019; Touati & Ollivier, 2021; Touati et al., 2023; Park et al., 2024), our approach harnesses SFs for IRL, aiming to match expected features of the expert. Within the body of work on imitation learning, SFs have been leveraged to pre-train behavior foundation models capable of rapid imitation (Pirotta et al., 2024) and within adversarial IRL typically serves as the basis for estimating the value function that best explains the expert (Lee et al., 2019; Filos et al., 2021; Abdulhai et al., 2022). In contrast, our work seeks to directly match SFs through a policy gradient update without requiring large, diverse datasets or costly bilevel optimization procedures.

## 3 PRELIMINARIES

**Reinforcement Learning (RL; Sutton & Barto, 2018)** considers a Markov Decision Process (MDP) defined by $\mathcal{M} = (\mathcal{S}, \mathcal{A}, P, r, \gamma, P_0)$, where $\mathcal{S}$ and $\mathcal{A}$ denote the state and action spaces, $P : \mathcal{S} \times \mathcal{A} \to \Delta(\mathcal{S})$ denotes the transition kernel, $r : \mathcal{S} \to \mathbb{R}$ is the reward function, $\gamma \in [0, 1)$ is the discount factor, and $P_0 \in \Delta(\mathcal{S})$ is the initial state distribution. Starting from the initial state $S_0 \sim P_0(\cdot)$ an agent takes actions according to its policy $\pi : \mathcal{S} \to \Delta(\mathcal{A})$ producing trajectories $\{S_0, A_1, S_1, \dots\} \sim \text{Pr}_\pi$ — the probability measure over trajectories where $S_0 \sim P_0$, $A_t \sim \pi(\cdot \mid S_t)$, and $S_{t+1} \sim P(\cdot \mid S_t, A_t)$. We write $\mathbb{E}_\pi$ to denote expectations with respect to states and actions sampled under $\text{Pr}_\pi$. The performance of a policy $\pi$ can be measured as the cumulative discounted sum of rewards obtained from an initial state, given by

$$J(\pi; r) = \mathbb{E}_{S \sim P_0(\cdot), A \sim \pi(\cdot|S)} \Big[ \underbrace{\mathbb{E}_\pi \Big[ \sum_{k=0}^\infty \gamma^k r(S_{t+k}) \mid S_t = S, A_t = A \Big]}_{Q_r^\pi(S, A)} \Big], \quad (1)$$

where $Q_r^\pi : \mathcal{S} \times \mathcal{A} \to \mathbb{R}$ is referred to as the action-value function. When the reward function is unambiguous, we write $J(\pi)$ and $Q^\pi$ in place of $J(\pi; r)$ and $Q_r^\pi$.

Successor Features (SF; Barreto et al., 2017; 2020) allow us to linearly factorize the action-value function as $Q_r^\pi(s, a) = \boldsymbol{\psi}^\pi(s, a)^\top w_r$ with the components: (1) $\boldsymbol{\psi}^\pi : \mathcal{S} \times \mathcal{A} \to \mathbb{R}^d$ being the expected discounted sum of *state features* $\boldsymbol{\psi}^\pi(s, a) = \mathbb{E}_\pi \big[ \sum_{k=0}^\infty \gamma^k \phi(S_{t+k}) \mid S_t = s, A_t = a \big]$ after applying the feature map $\phi : \mathcal{S} \to \mathbb{R}^d$, and (2) $w_r \in \mathbb{R}^d$ being a linear projection of the reward function $r$ onto the components of $\phi$ defined as $w_r = (\text{Cov}_\pi [\phi(S)])^{-1} \mathbb{E}_\pi [r(S) \phi(S)]$ (Touati & Ollivier, 2021). In practice, we can learn a parametric model $\boldsymbol{\psi}_\theta^\pi \approx \boldsymbol{\psi}^\pi$ via Temporal Difference (TD) learning (Sutton, 1988) by minimizing the following least-squares TD objective,

$$\mathcal{L}_{SF}(\theta; \bar{\theta}) = \mathbb{E}_{(S, A, S') \sim \mathcal{B}, A' \sim \pi(\cdot|S')} \Big[ \|\phi(S) + \gamma \boldsymbol{\psi}_{\bar{\theta}}^\pi(S', A') - \boldsymbol{\psi}_\theta^\pi(S, A)\|_2^2 \Big], \quad (2)$$

where the tuple $(S, A, S')$ is a state-action-next-state transition sampled from dataset $\mathcal{B}$. The parameters $\bar{\theta}$ denote the "target parameters" that are periodically updated from $\theta$ by either taking a direct copy or a moving average of $\theta$ (Mnih et al., 2015).

**Inverse Reinforcement Learning (IRL; Ng et al., 2000; Abbeel & Ng, 2004; Ziebart et al., 2008)** is the task of deriving behaviors using demonstrations through interacting with the environment. In contrast to RL where the agent improves its performance using the learned reward, Inverse

Reinforcement Learning (IRL) involves learning without access to the reward function; good performance is signalled by expert demonstrations. As highlighted in Swamy et al. (2021), this corresponds to minimizing an Integral Probability Metric (IPM) (Sun et al., 2019) between the agent's state-visitation occupancy and the expert's which is framed to minimize the imitation gap given by:

$$J(\pi_E) - J(\pi) \leq \sup_{f \in \mathcal{F}_\phi} \left( \mathbb{E}_\pi \Big[ \sum_{t=0}^\infty \gamma^t f(S_t) \Big] - \mathbb{E}_{\pi_E} \Big[ \sum_{t=0}^\infty \gamma^t f(S_t) \Big] \right) \tag{3}$$

where $\mathcal{F}_\phi : \mathcal{S} \to \mathbb{R}$ denotes the class of reward basis functions. Under this taxonomy, the agent being the minimization player selects a policy $\pi \in \Pi$ to compete with a discriminator that picks a reward moment function $f \in \mathcal{F}_\phi$ to maximize the imitation gap. This leads to a natural framing as a min-max game $\min_\pi \max_{f \in \mathcal{F}_\phi} J(\pi_E) - J(\pi)$.

By restricting the class of reward basis functions to be within span of some base-features $\phi$ such that $\mathcal{F}_\phi \in \{f(s) = \phi(s)^T w : \|w\|_2 \leq B\}$, the imitation gap becomes:

$$\begin{aligned}
J(\pi_E) - J(\pi) &\leq \sup_{\|w\|_2 \leq B} \mathbb{E}_{\pi_E} \Big[ \sum_{t=0}^\infty \gamma^t \phi(S_t)^\top w \Big] - \mathbb{E}_\pi \Big[ \sum_{t=0}^\infty \gamma^t \phi(S_t)^\top w \Big] \\
&= \sup_{\|w\|_2 \leq B} \left( \mathbb{E}_{S \sim P_0(\cdot), A \sim \pi_E(\cdot|S)} \big[ \boldsymbol{\psi}^{\pi_E}(S,A) \big] - \mathbb{E}_{S \sim P_0(\cdot), A \sim \pi(\cdot|S)} \big[ \boldsymbol{\psi}^\pi(S,A) \big] \right)^\top w,
\end{aligned} \tag{4}$$

where $\boldsymbol{\psi}^E(s,a)$ denotes the successor features of the expert policy $\pi_E$ for a given state $s$ and action $a$. Under this assumption, the agent that matches the successor features with the expert will minimize the performance gap across the class of restricted basis reward functions. A vector $w^\star$ optimizing the supremum in (4) is referred to as a *witness*—it describes a reward function that most clearly witnesses the distinction between $\pi$ and the expert demonstrations. Solving the objective of matching expected features between the agent and the expert has been studied in prior methods where previous approaches often resort to solving an adversarial game (Ziebart et al., 2008; Abbeel & Ng, 2004; Syed & Schapire, 2007; Syed et al., 2008). In the sequel, we introduce a non-adversarial approach that updates the policy to align the SFs between the expert and the agent, and does not require optimizing an explicit reward function to capture the behavioral divergence.

Naturally, the aforementioned assumption requires that $\phi$ induces a class $\mathcal{F}_\phi$ that is rich enough to contain the expert's underlying reward function. It is not generally possible to ensure this without privileged information—instead, we jointly *learn* features $\phi$ using recent advances in representation learning for RL (e.g., Farebrother et al., 2023; Park et al., 2024). We posit that such features, which are meant to distinguish between a diverse set of behaviors, will be rich enough to include the expert's reward in their span. Our experimental results validate that indeed this can be achieved.

## 4 SUCCESSOR FEATURE MATCHING

In this section, we describe Successor Feature Matching (SFM) — a state-only, non-adversarial algorithm for matching expected features between the agent and expert. The key concept underlying SFM is that, leveraging successor features, the *witness* $w$ in equation 4 can be approximated in closed form, yielding a reward function for RL policy optimization. Specifically, the difference in successor features between the agent and expert can itself act as the witness $w$—in this case, $w$ is parallel to the feature matching objective, implying that this witness maximally discriminates between the agent and expert's performance. Concretely, we have that,

$$w^\star_{\pi \to \pi_E} := B \frac{\widehat{\boldsymbol{\psi}}^E - \widehat{\boldsymbol{\psi}}^\pi}{\big\| \widehat{\boldsymbol{\psi}}^E - \widehat{\boldsymbol{\psi}}^\pi \big\|_2} \in \arg\max_{\|w\|_2 \leq B} \big( \widehat{\boldsymbol{\psi}}^E - \widehat{\boldsymbol{\psi}}^\pi \big)^\top w, \text{ where}$$

$$\widehat{\boldsymbol{\psi}}^E := \mathbb{E}_{S \sim P_0(\cdot),\, A \sim \pi_E(\cdot|S)} \big[ \boldsymbol{\psi}^E(S,A) \big] \quad \text{and} \quad \widehat{\boldsymbol{\psi}}^\pi := \mathbb{E}_{S \sim P_0(\cdot),\, A \sim \pi(\cdot|S)} \big[ \boldsymbol{\psi}^\pi(S,A) \big]. \tag{5}$$

Remarkably, this observation allows us to bypass the adversarial reward learning component of IRL by directly estimating the imitation-gap-maximizing reward function $r^\star_{\pi_\mu \to \pi_E}$ given by

$$r^\star_{\pi_\mu \to \pi_E}(x) = \phi(x)^\top w^\star_{\pi \to \pi_E} \propto \phi(x)^\top \big( \widehat{\boldsymbol{\psi}}^E - \widehat{\boldsymbol{\psi}}^\pi \big). \tag{6}$$

---

**Algorithm 1** Successor Feature Matching (SFM)

---

**Require:** Expert demonstrations $\tau^E = \{s_0^i, a_0^i, \ldots, s_{T-1}^i, a_{T-1}^i\}_{i=1}^M$
**Require:** Base feature loss $\mathcal{L}_{\text{feat}}$ and initialized parameters $\theta_{\text{feat}} = (\phi, f)$
**Require:** Initialized actor $\pi_\mu$, SF network and target $\boldsymbol{\psi}_\theta, \boldsymbol{\psi}_{\bar{\theta}}$, replay buffer $\mathcal{B}$
 1: **while** Training **do**
 2:     Rollout $\pi_\mu$ and add transitions to replay buffer $\mathcal{B}$
 3:     Update expected features of expert $\widehat{\boldsymbol{\psi}}^E$ with EMA using (7)
 4:     Sample independent minibatches $\mathcal{D}, \mathcal{D}' \subset \mathcal{S} \times \mathcal{A} \times \mathcal{S}$ from $\mathcal{B}$
 5:     Update SF network via $\nabla_\theta \mathbb{E}_{(S,A,S')\sim\mathcal{D}, A'\sim\pi_\mu(\cdot|S')} \big[ \|\phi(S) + \boldsymbol{\psi}_{\bar{\theta}}(S',A') - \boldsymbol{\psi}_\theta(S,A)\|_2^2 \big]$
 6:     Estimate witness $\hat{w} = \widehat{\boldsymbol{\psi}}^E - \widehat{\boldsymbol{\psi}}^\pi$ using Proposition 1 and minibatch $\mathcal{D}'$.
 7:     Update actor via $\nabla_\mu U(\pi_\mu; s \mapsto \phi(s)^\top \hat{w})$ using Proposition 2 and minibatch $\mathcal{D}$
 8:     Update base feature function via $\nabla_{\theta_{\text{feat}}} \mathcal{L}_{\text{feat}}(\theta_{\text{feat}})$
 9: **end while**

---

This insight enables us to replace the costly bi-level optimization of IRL in favor of solving a single RL problem. The remaining challenge, however, is determining *how* to estimate $w^\star_{\pi\to\pi_E}$.

A natural first step in estimating $w^\star_{\pi\to\pi_E}$ is to leverage the provided expert demonstrations consisting of $M$ trajectories $\{\tau^i = (s_1^i, \ldots, s_{T_i}^i)\}_{i=1}^M$. These demonstrations allow us to compute $\widehat{\boldsymbol{\psi}}^E$ as,

$$\widehat{\boldsymbol{\psi}}^E = \frac{1}{M} \sum_{i=1}^M \sum_{t=1}^{T_i} \gamma^{t-1} \phi(s_t^i). \tag{7}$$

A naïve approach might then attempt to estimate $\widehat{\boldsymbol{\psi}}^\pi$ from initial states—however, this proves to be challenging, as it precludes bootstrapped TD estimates, and Monte Carlo estimators have prohibitively high variance for $\gamma$ near 1. Instead, we leverage a key result that allows us to estimate this quantity more effectively by bootstrapping from arbitrary transitions from the environment.

**Proposition 1.** *Let $\mathcal{B}$ denote a buffer of trajectories sampled from arbitrary stationary Markovian policies in the given MDP with initial state distribution $P_0$. For any stochastic policy $\pi$,*

$$\widehat{\boldsymbol{\psi}}^\pi = (1-\gamma)^{-1} \mathbb{E}_{(S,S')\sim\mathcal{B}} \Big[ \mathbb{E}_{A\sim\pi(\cdot|S)} \big[ \boldsymbol{\psi}^\pi(S,A) \big] - \gamma \mathbb{E}_{A'\sim\pi(\cdot|S')} \big[ \boldsymbol{\psi}^\pi(S',A') \big] \Big]. \tag{8}$$

The proof of Proposition 1 is deferred to Appendix A. Notably, a key aspect of this estimator is its ability to use samples from a different state-visitation distribution, effectively allowing us to use a replay buffer $\mathcal{B}$ of the agent's experience to estimate $\widehat{\boldsymbol{\psi}}^\pi$.

With estimates of $\widehat{\boldsymbol{\psi}}^E$ and $\widehat{\boldsymbol{\psi}}^\pi$ in hand, one might be tempted to apply any RL algorithm to optimize $r^\star_{\pi\to\pi_E}$. However, this approach overlooks the structure provided by the learned successor features. Instead, we show how these features can be directly leveraged to derive a novel policy gradient method that optimizes the policy to directly match the difference in features.

### 4.1 A Policy Gradient Method for Successor Feature Matching

Instead of directly optimizing the reward function $r^\star_{\pi\to\pi_E}$, we now derive a policy gradient (Sutton et al., 1999) that directly aligning the successor features of the agent and expert. To this end, we leverage our learned successor features — which are already required for estimating the witness $w^\star_{\pi\to\pi_E}$ — to calculate the value-function under the reward $r^\star_{\pi\to\pi_E}$, eliminating the need for a separate critic to estimate $Q^\pi_{r^\star_{\pi\to\pi_E}}$. This defines the following off-policy policy gradient objective (Degris et al., 2012) for the policy $\pi_\mu$ parametrized by $\mu$:

$$U(\pi_\mu; r^\star_{\pi_\mu\to\pi_E}) = \mathbb{E}_{S\sim\rho^\beta, A\sim\pi_\mu(\cdot|S)} \big[ \underbrace{\boldsymbol{\psi}_\theta^{\pi_\mu}(S,A)^\top w^\star_{\pi_\mu\to\pi_E}}_{Q^{\pi_\mu}_{r^\star_{\pi_\mu\to\pi_E}}} \big], \tag{9}$$

where $\beta : \mathcal{S} \to \Delta(\mathcal{A})$ is a policy different from $\pi_\mu$. In practice, we can view $\rho^\beta$ as being a replay buffer $\mathcal{B}$ containing experience collected throughout training. Given this objective, we now derive the policy gradient with respect to $\mu$ in the following result with the proof given in Appendix A.

**Proposition 2.** *For stochastic policies $\pi : \mathcal{S} \to \Delta(\mathcal{A})$ the policy gradient under which the return most steeply increases for the reward function $r^\star_{\pi_\mu \to \pi_E}$ defined in equation 6 is given by,*

$$\nabla_\mu U(\pi_\mu; r^\star_{\pi_\mu \to \pi_E}) = \left(w^\star_{\pi_\mu \to \pi_E}\right)^\top \left(\mathbb{E}_{S \sim \rho^\beta, A \sim \pi_\mu(\cdot|S)}\left[\nabla_\mu \log \pi_\mu(A \mid S) \, \boldsymbol{\psi}^{\pi_\mu}_\theta(S, A)\right]\right) . \quad (10)$$

*Alternatively, for* deterministic *policies $\pi : \mathcal{S} \to \mathcal{A}$, the deterministic policy gradient (Silver et al., 2014) for the reward function $r^\star_{\pi_\mu \to \pi_E}$ defined in equation 6 is given by,*

$$\nabla_\mu U(\pi_\mu; r^\star_{\pi_\mu \to \pi_E}) = \left(w^\star_{\pi_\mu \to \pi_E}\right)^\top \left(\mathbb{E}_{S \sim \rho^\beta}\left[\nabla_\mu \pi_\mu(S) \, \nabla_A \boldsymbol{\psi}^{\pi_\mu}_\theta(S, A)\right]\right) . \quad (11)$$

From Proposition 2 we can see that the SFM policy gradient operates by directly optimizing the alignment between the agent's successor features and the expert's by changing the policy in the direction that best aligns $\boldsymbol{\psi}^{\pi_\mu}$ with $\boldsymbol{\psi}^E$. This approach simplifies policy optimization by leveraging the computed successor features to directly guide the alignment of the agent's feature occupancy with that of the expert. Furthermore, Proposition 2 along with Equation 2 provide drop-in replacements to the actor and critic losses found in many popular methods (e.g., Fujimoto et al., 2018; 2023; Haarnoja et al., 2018) allowing for the easy integration of SFM with most actor-critic methods.

The overall SFM policy gradient method, summarized in Algorithm 1, encompasses both estimating the witness $w^\star_{\pi \to \pi_E}$ as well as the policy optimization procedure to reduce the imitation gap using this witness. However, so far, we have assumed access to base features $\phi$ when estimating the successor features. In the following, we describe how these features, too, can be learned from data.

## 4.2 BASE FEATURE FUNCTION

We described in §3 that SFs depends on a base feature function $\phi : \mathcal{S} \to \mathbb{R}^d$. In this work, SFM learns the base features jointly while learning the policy. Base feature methods are parameterized by pairs $\theta_{\text{feat}} = (\phi, \mathsf{f})$ together with a loss $\mathcal{L}_{\text{feat}}$, where $\phi : \mathcal{S} \to \mathbb{R}^d$ is a state feature map, $\mathsf{f}$ is an auxiliary object that may be used to learn $\phi$, and $\mathcal{L}_{\text{feat}}$ is a loss function defined for $\phi$ and $\mathsf{f}$.

Before discussing the learning of base features, we note an important point: when we don't have access to expert actions, we can still compute $\widehat{\boldsymbol{\psi}}^\pi$ and the policy gradient by using action-independent base features. While our method can handle problems where rewards depend on both states and actions (requiring expert action labels), in many practical applications expert actions are unavailable. As our experiments in §5 demonstrate, SFM can effectively learn imitation policies without requiring access to expert actions substantially broadening the applicability of our approach.

Below, we briefly outline the base feature methods considered in this work.

**Random Features (Random)**: Here, $\phi$ is a randomly-initialized neural network, and $\mathsf{f}$ is discarded. The network $\phi$ remains fixed during training ($\mathcal{L}_{\text{feat}} \equiv 0$).

**Autoencoder Features (AE)**: Here, $\phi : \mathcal{S} \to \mathbb{R}^d$ compresses states to latents in $\mathbb{R}^d$, and $\mathsf{f} : \mathbb{R}^d \to \mathcal{S}$ tries to reconstruct the state from the latent. The loss $\mathcal{L}_{\text{feat}}$ is given by the AE loss $\mathcal{L}_{\text{AE}}$,

$$\mathcal{L}_{\text{AE}}(\theta_{\text{feat}}) = \mathbb{E}_{S \sim \mathcal{B}}\left[\|\mathsf{f}(\phi(S)) - S\|_2^2\right], \quad \theta_{\text{feat}} = (\phi, \mathsf{f}). \quad (12)$$

**Inverse Dynamics Model (IDM; Pathak et al., 2017)**: Here, $\mathsf{f} : \mathbb{R}^d \times \mathbb{R}^d \to \mathcal{A}$ is a function that tries to predict the action that lead to the transition between embeddings $\phi : \mathcal{S} \to \mathbb{R}^d$ of consecutive states. The loss $\mathcal{L}_{\text{feat}}$ is given by the IDM loss $\mathcal{L}_{\text{IDM}}$,

$$\mathcal{L}_{\text{IDM}}(\theta_{\text{feat}}) = \mathbb{E}_{(S, A, S') \sim \mathcal{B}}\left[\|\mathsf{f}(\phi(S), \phi(S')) - A\|_2^2\right], \quad \theta_{\text{feat}} = (\phi, \mathsf{f}). \quad (13)$$

**Forward Dynamics Model (FDM)**: Here, $\mathsf{f} : \mathbb{R}^d \times \mathcal{A} \to \mathcal{S}$ is a function that tries to predict the next state in the MDP given the embedding of the current state and the chosen action. The loss $\mathcal{L}_{\text{feat}}$ is given by the FDM loss $\mathcal{L}_{\text{FDM}}$,

$$\mathcal{L}_{\text{FDM}}(\theta_{\text{feat}}) = \mathbb{E}_{(S, A, S') \sim \mathcal{B}}\left[\|\mathsf{f}(\phi(S), A) - S'\|_2^2\right], \quad \theta_{\text{feat}} = (\phi, \mathsf{f}). \quad (14)$$

**Hilbert Representations (HR; Park et al., 2024)**: The feature map $\phi : \mathcal{S} \to \mathbb{R}^d$ of HR is meant to estimate a *temporal distance*: the idea is that the difference between state embeddings $f^*_\phi(s, g) =$

$\|\phi(s) - \phi(g)\|$ approximates the amount of timesteps required to traverse between the state $s \in \mathcal{S}$ and randomly sampled goal $g \in \mathcal{S}$. Here, f is discarded, and $\mathcal{L}_{\text{feat}}$ is the HR loss $\mathcal{L}_{\text{HR}}$,

$$\mathcal{L}_{\text{HR}}(\theta_{\text{feat}}) = \mathbb{E}_{(S,S') \sim \mathcal{B}, G \sim \mathcal{B}} \left[ \ell_\tau^2 \left( -\mathbb{1}(S \neq G) - \gamma \text{sg}\{f_\phi^*(S', G)\} + f_\phi^*(S, G) \right) \right], \ \theta_{\text{feat}} = (\phi, \emptyset),$$
(15)

where sg denotes the stop-gradient operator, $\gamma$ is the discount factor, and $\ell_\tau^2$ is the $\tau$-expectile loss (Newey & Powell, 1987), as a proxy for the max operator in the Bellman backup (Kostrikov et al., 2022). In practice, $\text{sg}\{f_\phi^*(S', G)\}$ is replaced by $f_{\bar{\phi}}^*(S', G)$, where $\bar{\phi}$ is a delayed *target network* tracking $\phi$, much like a target network in DQN (Mnih et al., 2015).

Finally, our framework does not preclude the use of adversarially-trained features, although we maintain that a key advantage of the framework is that it *does not require* adversarial training. To demonstrate the influence of such features, we consider training base features via IRL.

**Adversarial Representations (Adv):** The embedding $\phi : \mathcal{S} \to \mathbb{R}^d$ is trained to maximally distinguish the features on states visited by the learned policy from the expert policy. That is, $\mathcal{L}_{\text{feat}}$ is given by $\mathcal{L}_{\text{Adv}}$ which adversarially maximizes an imitation gap similar to equation 4,

$$\mathcal{L}_{\text{Adv}}(\theta_{\text{feat}}) = - \left\| \mathbb{E}_\pi \big[ \phi(S) \big] - \mathbb{E}_{\pi_E} \big[ \phi(S') \big] \right\|_2^2, \quad \theta_{\text{feat}} = (\phi, \emptyset).$$
(16)

In our experiments, we evaluated SFM with each of the base feature methods discussed above. A comparison of their performance is given in Figure 7. Our SFM method adapts familiar deterministic policy gradient algorithms, particularly TD3 (Fujimoto et al., 2018) and TD7 (Fujimoto et al., 2023), for policy optimization through the actor loss of equation 11. We further provide results for stochastic policy gradient methods in Appendix B.1. Full implementation details are provided in Appendix C, and we now demonstrate the performance of SFM in the following section.

## 5 EXPERIMENTS

Through our experiments, we aim to analyze (1) how well SFM performs relative to competing non-adversarial and state-only adversarial methods at imitation from a single expert demonstration, (2) the robustness of SFM and its competitors to their underlying policy optimizer, and (3) which features lead to strong performance for SFM. Our results are summarized in Figures 2, 4, and 7, respectively, and are discussed in the subsections below. Ultimately, our results confirm that SFM indeed outperforms its competitors, achieving state-of-the-art performance on a variety of **single-demonstration** tasks, even surpassing the performance of agents that have access to expert actions.

### 5.1 EXPERIMENTAL SETUP

We evaluate our method 10 environments from the DeepMind Control (DMC; Tunyasuvunakool et al., 2020) suite. Following the investigation in Jena et al. (2020) which showed that the IRL algorithms are prone to biases in the learned reward function, we consider infinite horizon tasks where all episodes are truncated after 1000 steps in the environment. For each task, we collected expert demonstrations by training a TD3 (Fujimoto et al., 2018) agent for 1M environment steps. In our experiments, the agent is provided with a single expert demonstration which is kept fixed during the training phase. The agents are trained for 1M environment steps and we report the mean performance across 10 seeds with 95% confidence interval shading following the best practices in RLiable (Agarwal et al., 2021). For the RLiable plots, we use the returns obtained by a random policy and the expert policy to compute the normalized returns. Our implementation of SFM is in Jax (Bradbury et al., 2018) and it takes about ~2.5 hours for one run on a single NVIDIA A100 GPU. We provide implementation details in Appendix C and hyperparameters in Appendix D.

**Baselines.** Our baselines include a state-only version of moment matching (MM; Swamy et al., 2021), which is an adversarial IRL approach where the integral probability metric (IPM) is replaced with the Jenson-Shannon divergence (which was shown to achieve better or comparable performance with GAIL (Swamy et al., 2022)). We implemented state-only MM by changing the discriminator network to depend only on the state and not on actions. Furthermore, we replace the RL optimizer in MM to TD7 (Fujimoto et al., 2023) to keep parity with SFM. We compare SFM to another state-only baseline GAIfO (Torabi et al., 2018) where the discriminator learns to distinguish between the state transitions of the expert and the agent. Since, to our knowledge, no official implementation of

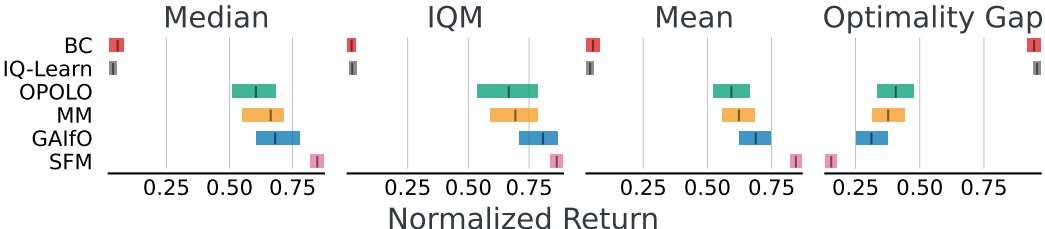

Figure 2: RLiable (Agarwal et al., 2021) plots of the proposed method SFM with an offline method BC, a non-adversarial method IQ-Learn that uses expert action labels and adversarial state-only methods: OPOLO, MM and GAIfO across 10 tasks from DMControl suite.

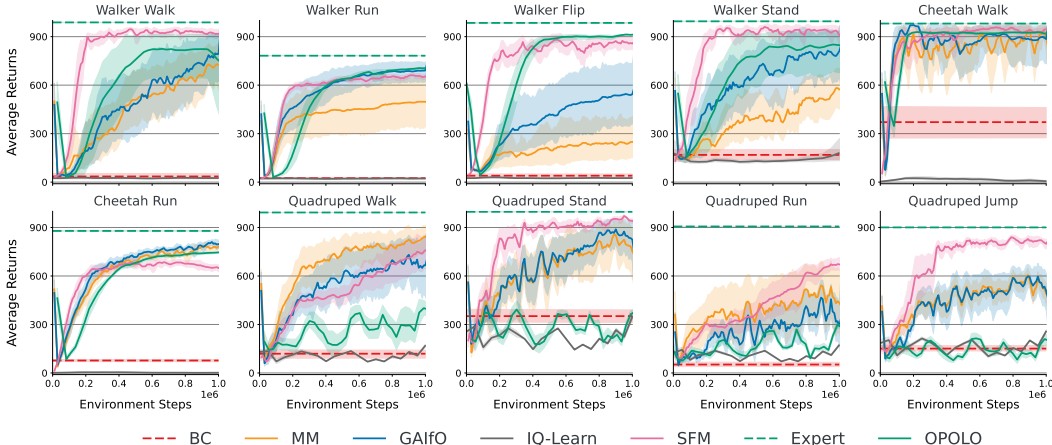

Figure 3: Per-task learning curves of IRL methods with the TD7 (Fujimoto et al., 2023) policy optimizer on single-demonstration imitation in DMC. Notably, IQ-Learn and BC require access to expert actions, while (state-only) MM, GAIfO, and SFM learn from expert states alone. Results are averaged across 10 seeds, and are shown with 95% confidence intervals.

GAIfO is available, we implemented our version of GAIfO with a similar architecture to the MM framework. Here, we replace their use of TRPO (Schulman et al., 2015) with either TD3 or TD7 to maintain parity. Additionally, for adversarial approaches, we impose a gradient penalty (Gulrajani et al., 2017) on the discriminator, learning rate decay, and Optimistic Adam (Daskalakis et al., 2018) to help stabilize training. We also compare with OPOLO (Zhu et al., 2020) and use the official implementation for our experiments. Apart from state-only adversarial approaches, we compare with behavior cloning (BC; Pomerleau, 1988) which is a supervised learning based imitation learning method trained to match actions taken by the expert. Lastly, we compare SFM with IQ-Learn (Garg et al., 2021) – a non-adversarial IRL algorithm which learns the Q-function using inverse Bellman operator (Piot et al., 2016). Notably, BC and IQ-Learn require the expert action labels.

## 5.2 RESULTS

**Quantitative Results** Figure 2 presents the RLiable plots (Agarwal et al., 2021) aggregated over DMC tasks. We observe that SFM learns to solve the task with a single demonstration and significantly outperforms the non-adversarial BC (Pomerleau, 1988) and IQ-Learn (Garg et al., 2021) baselines. Notably, SFM achieves this without using the action labels from the demonstrations. We believe behavior cloning (BC) fails in this regime of few expert demonstrations as the agent is unpredictable upon encountering states not in the expert dataset (Ross & Bagnell, 2010). We further observe that SFM outperforms our implementation of state-only adversarial baselines.Furthermore, SFM has a significantly lower optimality gap, indicating that the baselines are more likely to perform much worse than the expert. Among the state-only adversarial approaches, GAIfO leverages a more powerful discriminator based on the state transition as compared to only states used in MM and thereby performs better. To further analyze the gains, we report the average returns across each task

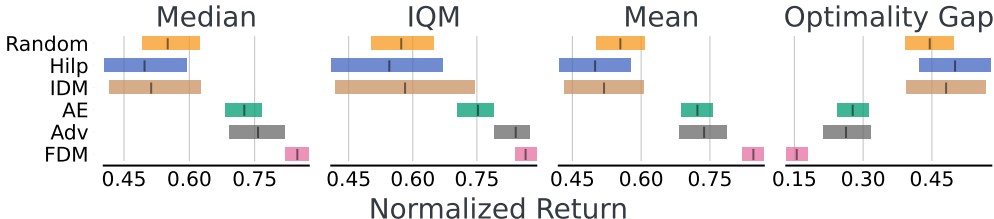

Figure 4: Performance of state-only IRL algorithms under the weaker TD3 policy optimizer.

Figure 5: Effect of different base features on the performance of SFM. Here, we compare with Random, Inverse Dynamics Model (IDM), Hilbert Representations (Hilp), Autoencoder (AE), Adversarial (Adv) and Forward Dynamics Models (FDM). FDM was found to work best across DMC tasks. Note that all base feature functions were jointly learned during training.

in Figure 3. We observe that OPOLO does well on walker and cheetah domains; however, it struggles with the quadruped domain where we posit its reliance on a learned inverse dynamics model becomes problematic due to the challenges of accurately modeling these more complex dynamics. We observe that SFM converges faster when compared to leading methods, suggesting improved sample efficiency relative to its competitors. SFM does not use techniques like gradient penalties which are often required when training adversarial methods (Swamy et al., 2021; Ren et al., 2024). Lastly, SFM outperforms MM and GAIfO on most tasks across the quadruped and walker domains.

**Robustness with weaker policy optimizers** In this work, the network architecture for SFM and the state-only baselines is inspired from TD7 (Fujimoto et al., 2023). TD7 is a recent algorithm presenting several tricks to attain improved performance relative to its celebrated predecessor TD3. To evaluate how robust these methods are to the quality of the RL algorithm, we also study the performance characteristics when employing the relatively weaker TD3 optimizer. The RLiable plots in Figure 4 present the efficacy of SFM to learn with the less performant TD3 optimizer. Remarkably, the performance of SFM (TD3) is similar to the SFM (Figure 2) demonstrating the efficacy of our non-adversarial method to learn with other off-the-shelf RL algorithms. However, the adversarial baselines did not perform as well when deployed with TD3. To further understand the performance difference, in Figure 6 we see that SFM attains significant performance gains across tasks in the quadruped domain. In contrast, the adversarial state-only baselines perform similarly on tasks in the cheetah and walker domains for both RL optimizers.

**Ablation over base features** In Figure 7, we study the performance of SFM with various base feature $\phi$. As discussed in §4.2, we experiment with Random Features, Inverse Dynamics Models (IDM; Pathak et al., 2017), Hilbert Representations (Hilp; Park et al., 2024), Forward Dynamics Model (FDM), Autoencoder (AE), and Adversarial (Adv) features. Through our experiments, we observe that FDM achieves superior results when compared with other base feature functions (Figure 5). In Figure 7 and Table 3, we see that, IDM features performed similarly to FDM on walker and cheetah domains, but did not perform well on quadruped tasks. We believe it is challenging to learn IDM features on quadruped domain which has been observed in prior works (Park et al., 2024; Touati et al., 2023). Similar trends were observed for Hilp features and we suspect that learning the notion of temporal distance during online learning is challenging as the data distribution changes while training. Random features performed well on quadruped domain, but did not perform well on cheetah and walker tasks. Autoencoder (AE) and adversarial (Adv) features did well across RLiable metrics, however FDM features achieved better performance– we suspect that leveraging structure from the dynamics leads to superior performance. Moreover, learning adversarial features required tricks like gradient penalty and learning rate decay. We believe SFM can leverage any representa-

tion learning technique to obtain base features and a potential avenue for future work is to leverage pretrained features for more complex tasks to speed up learning.

## 6 DISCUSSION

We introduced SFM—a novel non-adversarial method for IRL that requires no expert action labels. Our method learns to match the expert's successor features, derived from adaptively learned base features, using direct policy search as opposed to solving a minmax adversarial game. Through experiments on several standard imitation learning benchmarks, we have shown that state-of-the-art imitation is achievable with a non-adversarial approach, thereby providing an affirmative answer to our central research question.

Consequently, SFM is no less stable to train than its online RL subroutine. This is not the case with adversarial methods, which involve complex game dynamics during training. Much like the rich literature on GANs (Goodfellow et al., 2014; Gulrajani et al., 2017; Kodali et al., 2018), adversarial IRL methods often require several tricks to stabilize the optimization, such as gradient penalties, bespoke optimizers, and careful hyperparameter tuning.

Beyond achieving state-of-the-art performance, SFM demonstrated an unexpected feat: it is exceptionally robust to the policy optimization subroutine. Notably, when using the weaker TD3 policy optimizer, SFM performs almost as well as it does with the relatively stronger TD7 optimizer. This is in stark contrast to the baseline methods, which performed considerably worse under the weaker policy optimizer. As such, we expect that SFM can be broadly useful and easier to deploy on resource-limited systems, which is often a constraint in robotics applications.

Interestingly, SFM follows a recent trend in model alignment that foregoes explicit reward modeling for direct policy search. This was famously exemplified in RLHF with DPO (Rafailov et al., 2024) and its subsequent extensions (Azar et al., 2024; Munos et al., 2024). It is worth noting that SFM, unlike DPO, *does* require modeling state features. However, the state features modeled by SFM are *task-agnostic*, and we found in particular that state embeddings for latent dynamics models suffice. We emphasize that this is a reflection of the more complicated dynamics inherent to general RL problems, unlike natural language problems which have trivial dynamics.

SFM is not the first non-adversarial IRL method; we note that IQ-Learn (Garg et al., 2021) similarly reduces IRL to RL. However, we showed that SFM substantially outperforms IQ-Learn in practice, and more importantly, it does so *without access to expert action labels*. Indeed, to our knowledge, SFM is *the first* non-adversarial state-only interactive IRL method. This opens the door to exciting possibilities, such as imitation learning from video and motion-capture data, which would not be possible for methods that require knowledge of the expert's actions. We believe that the simpler, non-adversarial nature of SFM training will be highly useful for scaling to such problems.

**Limitations**     While SFM is simpler than IRL methods, it still doesn't theoretically alleviate the exploration problem that IRL methods encounter. A promising direction of future work could combine SFM with resets (Swamy et al., 2023) or hybrid IRL (Ren et al., 2024) to improve sample efficiency. Alternatively, SFM can leverage existing exploration algorithms with certain methods leveraging successor features being particularly amenable (e.g., Machado et al., 2020).

## ACKNOWLEDGEMENTS

The authors would like to thank Lucas Lehnert, Adriana Hugessen, Gokul Swamy, Juntao Ren, Andrea Tirinzoni, and Ahmed Touati for their valuable feedback and discussions. The writing of the paper benefited from discussions with Darshan Patil, Mandana Samiei, Matthew Fortier, Zichao Li and anonymous reviewers. This work was supported by Fonds de Recherche du Québec, National Sciences and Engineering Research Council of Canada (NSERC), Calcul Québec, Canada CIFAR AI Chair program, and Canada Excellence Research Chairs (CERC) program. The authors are also grateful to Mila (mila.quebec) IDT and Digital Research Alliance of Canada for computing resources. Sanjiban Choudhury is supported in part by Google Google Faculty Research Award, OpenAI SuperAlignment Grant, ONR Young Investigator Award, NSF RI #2312956, and NSF FRR#2327973.

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

## A PROOFS

Before proving Proposition 1, we begin by proving some helpful lemmas. First, we present a simple generalization of a result from Garg et al. (2021).

**Lemma 1.** *Let $\mu$ denote any discounted state-action occupancy measure for an MDP with state space $\mathcal{S}$ and initial state distribution $P_0$, and let $\mathcal{V}$ denote a vector space. Then for any $f : \mathcal{S} \to \mathcal{V}$, the following holds,*

$$\mathbb{E}_{(S,A)\sim\mu}\left[f(S) - \gamma\mathbb{E}_{S'\sim P(\cdot|S,A)}[f(S')]\right] = (1-\gamma)\mathbb{E}_{S\sim P_0}\left[f(S)\right].$$

*Proof.* Firstly, any discounted state-action occupancy measure $\mu$ is identified with a unique policy $\pi^\mu$ as shown by Ho & Ermon (2016). So, $\mu$ is characterized by

$$\mu(\mathrm{d}s\mathrm{d}a) = (1-\gamma)\pi^\mu(\mathrm{d}a \mid s)\sum_{t=0}^{\infty}\gamma^t p_t^\mu(\mathrm{d}s),$$

where $p_t^\mu(S) = \mathrm{Pr}_{\pi^\mu}(S_t \in S)$ is the state-marginal distribution under policy $\pi^\mu$ at timestep $t$. Expanding the LHS of the proposed identity yields

$$\begin{aligned}
&\mathbb{E}_{(S,A)\sim\mu}\left[f(S) - (1-\gamma)\gamma\mathbb{E}_{S'\sim P(\cdot|S,A)}[f(S')]\right]\\
&= (1-\gamma)\sum_{t=0}^{\infty}\gamma^t\mathbb{E}_{S\sim p_t^\mu}[f(S)] - \gamma\mathbb{E}_{(S,A)\sim\mu}\mathbb{E}_{S'\sim P(\cdot|S,A)}[f(S')]\\
&= (1-\gamma)\sum_{t=0}^{\infty}\gamma^t\mathbb{E}_{S\sim p_t^\mu}[f(S)] - (1-\gamma)\sum_{t=0}^{\infty}\gamma^{t+1}\mathbb{E}_{S\sim p_t^\mu}\mathbb{E}_{A\sim\pi^\mu(\cdot|S)}\mathbb{E}_{S'\sim P(\cdot|S,A)}[f(S')]\\
&= (1-\gamma)\sum_{t=0}^{\infty}\gamma^t\mathbb{E}_{S\sim p_t^\mu}[f(S)] - (1-\gamma)\sum_{t=0}^{\infty}\gamma^{t+1}\mathbb{E}_{S\sim p_{t+1}^\mu}[f(S)]\\
&= (1-\gamma)\sum_{t=0}^{\infty}\gamma^t\mathbb{E}_{S\sim p_t^\mu}[f(S)] - (1-\gamma)\sum_{t=1}^{\infty}\gamma^t\mathbb{E}_{S\sim p_t^\mu}[f(S)]\\
&= (1-\gamma)\mathbb{E}_{S\sim P_0}[f(S)],
\end{aligned}$$

since $p_0^\mu = P_0$ (the initial state distribution) for any $\mu$. $\qquad\square$

Intuitively, we will invoke Lemma 1 with $f$ denoting the successor features to derive an expression for the initial state successor features via state transitions sampled from a replay buffer.

**Proposition 1.** *Let $\mathcal{B}$ denote a buffer of trajectories sampled from arbitrary stationary Markovian policies in the given MDP with initial state distribution $P_0$. For any stochastic policy $\pi$,*

$$\widehat{\boldsymbol{\psi}}^\pi = (1-\gamma)^{-1}\mathbb{E}_{(S,\,S')\sim\mathcal{B}}\left[\mathbb{E}_{A\sim\pi(\cdot|S)}\left[\boldsymbol{\psi}^\pi(S,A)\right] - \gamma\mathbb{E}_{A'\sim\pi(\cdot|S')}\left[\boldsymbol{\psi}^\pi(S',A')\right]\right]. \qquad (8)$$

*Proof.* Suppose $\mathcal{B}$ contains rollouts from policies $\{\pi_k\}_{k=1}^N$ for some $N \in \mathbb{N}$. Each of these policies $\pi_k$ induces a discounted state-action occupancy measure $\mu_k$. Since the space of all discounted state-action occupancy measures is convex (Dadashi et al., 2019), it follows that $\mu = \frac{1}{N}\sum_{k=1}^N\mu_k$ is itself a discounted state-action occupancy measure.

Consider the function $f : \mathcal{S} \to \mathbb{R}^d$ given by $f(s) = \mathbb{E}_{A\sim\pi(\cdot|s)}\boldsymbol{\psi}^\pi(s,A)$. We have

$$\mathbb{E}_{(S,S')\sim\mathcal{B}}[f(S) - \gamma f(S')]$$
$$= \mathbb{E}_{k\sim\mathsf{Uniform}(\{1,\dots,N\})}\mathbb{E}_{(S,A)\sim\mu_k, S'\sim P(\cdot|S,A)}[f(S) - \gamma f(S')]$$
$$= \mathbb{E}_{(S,A)\sim\mu, S'\sim P(\cdot|S,A)}[f(S) - \gamma f(S')]$$
$$= \mathbb{E}_{(S,A)\sim\mu}\left[f(S) - \gamma\mathbb{E}_{S'\sim P(\cdot|S,A)}[f(S')]\right]$$
$$= (1-\gamma)\mathbb{E}_{S\sim P_0}[f(S)],$$

where the final step invokes Lemma 1, which is applicable since $\mu$ is a discounted state-action occupancy measure. The claim then follows by substituting $f(S)$ for $\mathbb{E}_{A\sim\pi(\cdot|S)}\boldsymbol{\psi}^\pi(S, A)$ and multipying through by $(1-\gamma)^{-1}$. $\qquad\square$

**Proposition 2.** *For stochastic policies $\pi : \mathcal{S} \to \Delta(\mathcal{A})$ the policy gradient under which the return most steeply increases for the reward function $r^\star_{\pi_\mu \to \pi_E}$ defined in equation 6 is given by,*

$$\nabla_\mu U(\pi_\mu; r^\star_{\pi_\mu\to\pi_E}) = \left(w^\star_{\pi_\mu\to\pi_E}\right)^\top \left(\mathbb{E}_{S\sim\rho^\beta, A\sim\pi_\mu(\cdot|S)}\left[\nabla_\mu\log\pi_\mu(A\mid S)\,\boldsymbol{\psi}^{\pi_\mu}_\theta(S,A)\right]\right). \quad (10)$$

*Alternatively, for deterministic policies $\pi : \mathcal{S} \to \mathcal{A}$, the deterministic policy gradient (Silver et al., 2014) for the reward function $r^\star_{\pi_\mu \to \pi_E}$ defined in equation 6 is given by,*

$$\nabla_\mu U(\pi_\mu; r^\star_{\pi_\mu\to\pi_E}) = \left(w^\star_{\pi_\mu\to\pi_E}\right)^\top \left(\mathbb{E}_{S\sim\rho^\beta}\left[\nabla_\mu\pi_\mu(S)\,\nabla_A\boldsymbol{\psi}^{\pi_\mu}_\theta(S,A)\right]\right). \quad (11)$$

*Proof.* That $\nabla_\mu U(\pi_\mu; r^\star_{\pi_\mu\to\pi_E})$ is the direction that most steeply increases the return under the reward function $r^\star_{\pi_\mu\to\pi_E}$ is established in Degris et al. (2012). It remains to derive an expression for $\nabla_\mu U(\pi_\mu; r^\star_{\pi_\mu\to\pi_E})$. Since $r^\star_{\pi_\mu\to\pi_E}$ is linear in the base features, given an estimate $\boldsymbol{\psi}_\theta$ of the successor features for policy $\pi_\mu$, the action-value function is given by

$$Q_\theta(s, a) = \boldsymbol{\psi}_\theta(s, a)^\top w^\star_{\pi_\mu\to\pi_E}.$$

Then, we have that

$$\begin{aligned}
\nabla_\mu U(\pi_\mu; r^\star_{\pi_\mu\to\pi_E}) &= \mathbb{E}_{S\sim\rho^\beta}\nabla_\mu\int_{\mathcal{A}}\pi_\mu(a\mid S)Q_\theta(S,a) \\
&= \mathbb{E}_{S\sim\rho^\beta}\nabla_\mu\int_{\mathcal{A}}\pi_\mu(a\mid S)\boldsymbol{\psi}_\theta(S,a)^\top w^\star_{\pi_\mu\to\pi_E}.
\end{aligned} \quad (17)$$

When $\pi_\mu$ is stochastic, the log-derivative trick yields

$$\begin{aligned}
\nabla_\mu U(\pi_\mu; r^\star_{\pi_\mu\to\pi_E}) &= \mathbb{E}_{S\sim\rho^\beta}\mathbb{E}_{A\sim\pi(\cdot|S)}\left[\nabla_\mu\log\pi_\mu(A\mid S)\boldsymbol{\psi}_\theta(S,A)^\top w^\star_{\pi_\mu\to\pi_E}\right] \\
&= (w^\star_{\pi_\mu\to\pi_E})^\top\left(\mathbb{E}_{S\sim\rho^\beta}\mathbb{E}_{A\sim\pi(\cdot|S)}\left[\nabla_\mu\log\pi_\mu(A\mid S)\boldsymbol{\psi}_\theta(S,A)\right]\right).
\end{aligned}$$

Alternatively, for deterministic policies $\pi_\mu$ (where, with notational abuse, we write $\pi_\mu(s) \in \mathcal{A}$), the deterministic policy gradient theorem (Silver et al., 2014) gives

$$\begin{aligned}
\nabla_\mu U(\pi_\mu; r^\star_{\pi_\mu\to\pi_E}) &= \mathop{\mathbb{E}}_{S\sim\rho^\beta}\left[\nabla_\mu\pi_\mu(S)\nabla_a[\boldsymbol{\psi}_\theta(S,a)^\top w^\star_{\pi_\mu\to\pi_E}]|_{a=\pi_\mu(S)}\right] \\
&= \mathop{\mathbb{E}}_{S\sim\rho^\beta}\left[\nabla_\mu\pi_\mu(S)\nabla_a\boldsymbol{\psi}_\theta(S,a)^\top|_{a=\pi_\mu(S)}w^\star_{\pi_\mu\to\pi_E}\right] \\
&= (w^\star_{\pi_\mu\to\pi_E})^\top\left(\mathop{\mathbb{E}}_{S\sim\rho^\beta}\left[\nabla_\mu\pi_\mu(S)\nabla_a\boldsymbol{\psi}_\theta(S,a)^\top|_{a=\pi_\mu(S)}w^\star_{\pi_\mu\to\pi_E}\right]\right).
\end{aligned}$$

as claimed.

$\qquad\square$

# B EXTENDED RESULTS

In this section, we provide the tables with average returns across tasks from DMControl Suite (Table 1, 2 & 3) and per-environment training runs for our study with weak policy optimizers and base feature functions ((Fig. 6 & 7).

| Task | BC | IQ-Learn | OPOLO | MM | GAIfO | SFM |
|---|---|---|---|---|---|---|
| Cheetah Run | $77.0 \pm 11.1$ | $1.4 \pm 1.4$ | $747.5 \pm 6.7$ | $\mathbf{781.6 \pm 30.7}$ | $777.2 \pm 45.0$ | $648.8 \pm 35.9$ |
| Cheetah Walk | $371.1 \pm 163.3$ | $5.7 \pm 6.9$ | $919 \pm 26.3$ | $895.6 \pm 128.2$ | $885.2 \pm 236.2$ | $\mathbf{945.1 \pm 33.9}$ |
| Quadruped Jump | $150.8 \pm 29.7$ | $260.7 \pm 12.0$ | $198.9 \pm 50.6$ | $489.4 \pm 104.6$ | $505.8 \pm 192.3$ | $\mathbf{799.1 \pm 47.8}$ |
| Quadruped Run | $52.1 \pm 22.4$ | $174.7 \pm 7.7$ | $291.5 \pm 43.9$ | $433.9 \pm 347.4$ | $289.4 \pm 227.3$ | $\mathbf{671.7 \pm 65.9}$ |
| Quadruped Stand | $351.6 \pm 68.5$ | $351.1 \pm 25.7$ | $378.7 \pm 37.3$ | $752.2 \pm 271.9$ | $804.7 \pm 211.5$ | $\mathbf{941.6 \pm 25.7}$ |
| Quadruped Walk | $119.0 \pm 40.9$ | $171.6 \pm 11.2$ | $391.8 \pm 57.9$ | $\mathbf{844.7 \pm 138.7}$ | $656.1 \pm 321.2$ | $759.9 \pm 177.5$ |
| Walker Flip | $39.6 \pm 17.7$ | $25.0 \pm 2.2$ | $\mathbf{913.6 \pm 2.5}$ | $249.1 \pm 230.8$ | $544.0 \pm 313.1$ | $856.9 \pm 64.5$ |
| Walker Run | $24.5 \pm 6.1$ | $22.4 \pm 1.6$ | $\mathbf{706.2 \pm 7.9}$ | $496.8 \pm 264.3$ | $690.7 \pm 101.9$ | $653.6 \pm 26.7$ |
| Walker Stand | $168.5 \pm 48.9$ | $181.1 \pm 135.8$ | $846.2 \pm 256.9$ | $574.2 \pm 209.3$ | $810.4 \pm 250.3$ | $\mathbf{909.4 \pm 96.9}$ |
| Walker Walk | $35.1 \pm 29.6$ | $25.3 \pm 2.6$ | $738.9 \pm 399.7$ | $725.3 \pm 234.8$ | $792.8 \pm 242.2$ | $\mathbf{916.5 \pm 43.4}$ |

Table 1: Returns achieved by BC, IQ-Learn, OPOLO, state-only MM, GAIfO and SFM across tasks on the DMControl Suite. The average returns and standard deviation across 10 seeds are reported.

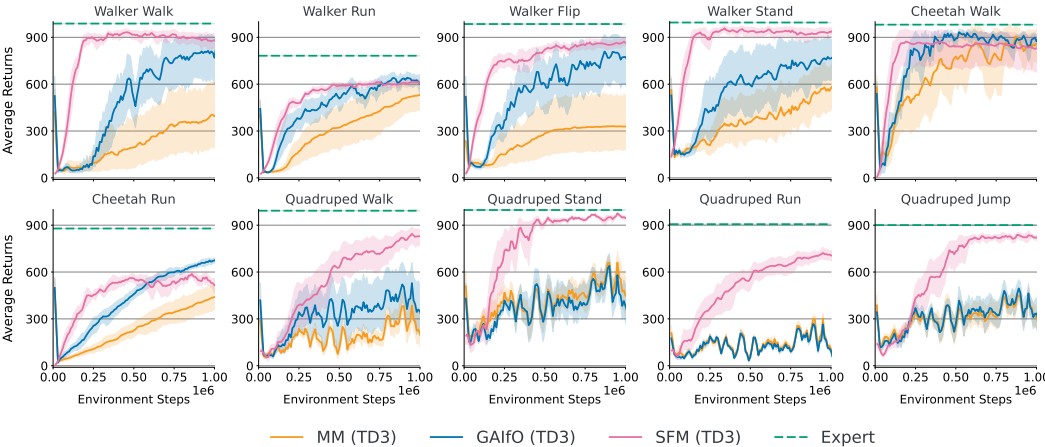

Figure 6: Comparison of state-only IRL methods using the weaker TD3 policy optimizer. Notably, only SFM consistently maintains strong performance with the weaker policy optimizer.

| Environment | MM (TD3) | GAIfO (TD3) | SFM (TD3) |
|---|---|---|---|
| Cheetah Run | $439.6 \pm 138.6$ | $\mathbf{674.0 \pm 27.7}$ | $514.7 \pm 77.9$ |
| Cheetah Walk | $859.6 \pm 165.3$ | $\mathbf{873.9 \pm 58.2}$ | $829.7 \pm 226.3$ |
| Quadruped Jump | $308.8 \pm 115.9$ | $334.3 \pm 159.8$ | $\mathbf{821.6 \pm 27.5}$ |
| Quadruped Run | $107.0 \pm 22.8$ | $94.4 \pm 23.1$ | $\mathbf{705.6 \pm 57.3}$ |
| Quadruped Stand | $449.7 \pm 206.1$ | $381.8 \pm 216.0$ | $\mathbf{946.3 \pm 20.5}$ |
| Quadruped Walk | $201.0 \pm 175.8$ | $347.3 \pm 246.4$ | $\mathbf{829.3 \pm 86.9}$ |
| Walker Flip | $328.7 \pm 287.8$ | $774.4 \pm 276.1$ | $\mathbf{865.5 \pm 37.7}$ |
| Walker Run | $530.2 \pm 163.1$ | $600.9 \pm 105.0$ | $\mathbf{606.5 \pm 30.2}$ |
| Walker Stand | $575.8 \pm 245.8$ | $764.8 \pm 220.7$ | $\mathbf{934.0 \pm 49.6}$ |
| Walker Walk | $395.1 \pm 351.1$ | $769.7 \pm 258.1$ | $\mathbf{880.1 \pm 75.8}$ |

Table 2: Comparison of state-only IRL methods using the weaker TD3 policy optimizer. This table presents returns achieved by state-only MM (TD3), GAIfO (TD3) and SFM (TD3) across tasks on the DMControl Suite. The average returns and standard deviation across 10 seeds are reported.

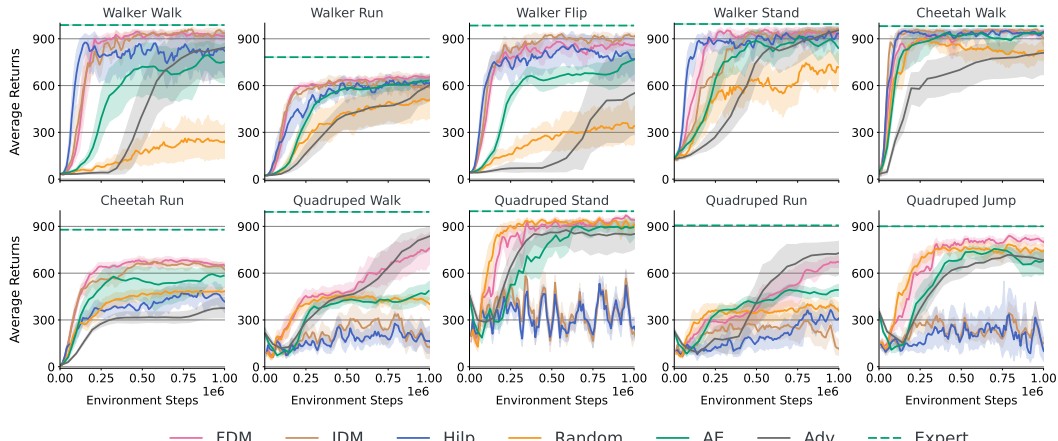

Figure 7: Effect of different base feature functions on the performance of the agent. Here, we compare with Random, Inverse Dynamics Model (IDM) (Pathak et al., 2017), Hilbert Representations (Hilp) (Park et al., 2024), Autoencoder (AE), Adversarial representations (Adv) and Forward Dynamics Models (FDM). FDM was found to work best across DMC tasks. Note that all base feature functions were jointly learned during training.

| Environment | Random | AE | Hilp | IDM | FDM | Adv |
|---|---|---|---|---|---|---|
| Cheetah Run | $484.4 \pm 45.4$ | $585.7 \pm 93.6$ | $417.8 \pm 118.8$ | $622.0 \pm 69.5$ | $\mathbf{648.8 \pm 35.9}$ | $374.1 \pm 113.7$ |
| Cheetah Walk | $823.8 \pm 107.6$ | $\mathbf{938.4 \pm 18.8}$ | $944.5 \pm 18.6$ | $908.2 \pm 88.4$ | $\mathbf{945.1 \pm 33.9}$ | $812.5 \pm 244.5$ |
| Quadruped Jump | $744.4 \pm 79.0$ | $678.0 \pm 126.6$ | $101.5 \pm 78.5$ | $151.3 \pm 59.0$ | $\mathbf{799.1 \pm 47.8}$ | $679.7 \pm 144.1$ |
| Quadruped Run | $356.8 \pm 92.7$ | $493.4 \pm 57.6$ | $311.8 \pm 94.5$ | $118.0 \pm 86.1$ | $671.7 \pm 65.9$ | $\mathbf{737.0 \pm 196.8}$ |
| Quadruped Stand | $914.0 \pm 33.1$ | $895.3 \pm 94.2$ | $259.1 \pm 102.0$ | $222.0 \pm 63.3$ | $\mathbf{941.6 \pm 25.7}$ | $858.4 \pm 140.3$ |
| Quadruped Walk | $402.8 \pm 68.7$ | $489.7 \pm 68.4$ | $166.0 \pm 138.8$ | $129.0 \pm 145.6$ | $759.9 \pm 177.5$ | $\mathbf{849.1 \pm 140.8}$ |
| Walker Flip | $341.8 \pm 204.4$ | $765.5 \pm 101.8$ | $771.9 \pm 197.0$ | $\mathbf{912.8 \pm 36.9}$ | $856.9 \pm 64.5$ | $565.1 \pm 439.9$ |
| Walker Run | $506.6 \pm 181.8$ | $632.2 \pm 41.7$ | $615.7 \pm 53.8$ | $589.5 \pm 139.0$ | $\mathbf{653.6 \pm 26.7}$ | $620.5 \pm 158.2$ |
| Walker Stand | $715.9 \pm 160.9$ | $836.2 \pm 140.0$ | $934.8 \pm 42.8$ | $\mathbf{960.6 \pm 22.9}$ | $909.4 \pm 96.9$ | $\mathbf{964.2 \pm 31.8}$ |
| Walker Walk | $243.8 \pm 189.7$ | $752.5 \pm 164.9$ | $821.8 \pm 242.5$ | $\mathbf{936.3 \pm 48.8}$ | $916.5 \pm 43.4$ | $849.6 \pm 289.8$ |

Table 3: Effect of different base feature functions on the performance of the agent. Here, we compare with Random, Inverse Dynamics Model (IDM) (Pathak et al., 2017), Hilbert Representations (Hilp) (Park et al., 2024), Autoencoder (AE), Adversarial representations (Adv), and Forward Dynamics Models (FDM). The table reports the returns achieved by each base feature function when trained with SFM across tasks on the DMControl Suite. The average returns and standard deviation across 10 seeds are reported. FDM was found to work best across DMC tasks. Note that all base feature functions were jointly learned during training.

## B.1 SFM WITH STOCHASTIC POLICY

To extend SFM to stochastic policies, we propose having an agent comprising of a stochastic actor parameterized to predict the mean and standard deviation of a multi-variate gaussian distribution. Here, for a given state $s$, the action is sampled using $a \sim \pi_\mu(\cdot|s) = \mathcal{N}(m_\mu(s), \text{diag}(\sigma_\mu^2(s)))$, where $m_\mu : \mathcal{S} \to \mathbb{R}^n$ and $\sigma_\mu^2 : \mathcal{S} \to \mathbb{R}_+^n$, for $\mathcal{A} = \mathbb{R}^n$. The SF network architecture $\psi_\theta$ is the same as the SFM (TD3) variant, where the network estimates the SF for a state-action pair. The SF-network can be updated using 1-step TD error using the base features of the current state similar to equation 2. The actor is updated using the update rule described in Proposition 2 where we estimate the policy gradient via reparameterization trick with Gaussian policies– particularly, we consider the class of Gaussian policies with diagonal covariance as in Haarnoja et al. (2018). Finally, to prevent the policy from quickly collapsing to a nearly-deterministic one, we also include a policy entropy bonus in our actor updates. We conduct experiments over the tasks from DMControl suite and present environment plots in Figure 8 and returns achieved in Table 4. We provide the implementations of SFM with stochastic policy in Appendix C.3

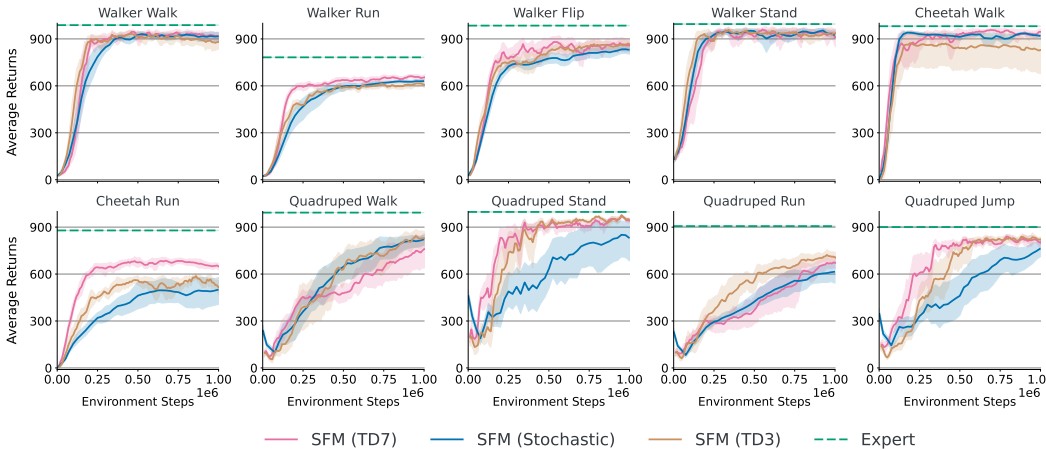

Figure 8: Comparison of variants of SFM with TD7, TD3 and an entropy regularized stochastic policy. We observe that SFM can be trained with stochastic polices. However, the variants with deterministic policy optimizers work better on some tasks than the stochastic policy.

| Environment | SFM (TD7) | SFM (TD3) | SFM (Stochastic) |
|---|---|---|---|
| Cheetah Run | $648.8 \pm 35.9$ | $514.7 \pm 77.9$ | $500.9 \pm 136.3$ |
| Cheetah Walk | $945.1 \pm 33.9$ | $829.7 \pm 226.3$ | $918.8 \pm 21.7$ |
| Quadruped Jump | $799.1 \pm 47.8$ | $821.6 \pm 27.5$ | $764.0 \pm 84.8$ |
| Quadruped Run | $671.7 \pm 65.9$ | $705.6 \pm 57.3$ | $614.9 \pm 113.4$ |
| Quadruped Stand | $941.6 \pm 25.7$ | $946.3 \pm 20.5$ | $829.5 \pm 224.1$ |
| Quadruped Walk | $759.9 \pm 177.5$ | $829.3 \pm 86.9$ | $821.9 \pm 56.3$ |
| Walker Flip | $856.9 \pm 64.5$ | $865.5 \pm 37.7$ | $830.4 \pm 45.4$ |
| Walker Run | $653.6 \pm 26.7$ | $606.5 \pm 30.2$ | $630.6 \pm 17.5$ |
| Walker Stand | $909.4 \pm 96.9$ | $934.0 \pm 49.6$ | $925.7 \pm 38.9$ |
| Walker Walk | $916.5 \pm 43.4$ | $880.1 \pm 75.8$ | $916.3 \pm 43.9$ |

Table 4: Comparison of SFM with a stochastic policy and variants based on deterministic policy optimizers (TD3 & TD7). The table reports the returns achieved by each base feature function when trained with SFM across tasks on the DMControl Suite. The average returns and standard deviation across 10 seeds are reported.

## C  IMPLEMENTATION DETAILS

Since SFM does not involve learning an explicit reward function and cannot leverage an off-the-shelf RL algorithm to learn a Q-funtion, we propose a novel architecture for our method. SFM is composed of 3 different components- actor $\pi_\mu$, successor features (SF) network $\boldsymbol{\psi}_\theta$, base feature function $\phi$ and $f$. Taking inspiration from state-of-the-art RL algorithms, we maintains target networks. Since, SF network acts similarly to a critic in actor-critic like algorithms, SFM comprises of two networks to estimate the SF (Fujimoto et al., 2018). Here, instead taking a minimum over estimates of SF from these two networks, our method performed better with average over the two estimates of SF. To implement the networks of SFM, we incorporated several components from the TD7 (Fujimoto et al., 2023), TD3 (Fujimoto et al., 2018), and SAC (Haarnoja et al., 2018) algorithm which are described in this section. Moreover, SFM does not require techniques like gradient penalty (Gulrajani et al., 2017), the OAdam optimizer (Daskalakis et al., 2018) and a learning rate scheduler.

### C.1  TD7-BASED NETWORK ARCHITECTURE

The architecture used in this work is inspired from the TD7 (Fujimoto et al., 2023) algorithms for continuous control tasks (Pseudo 1). We will describe the networks and sub-components used below:

- Two functions to estimate the SF ($\boldsymbol{\psi}_{\theta_1}, \boldsymbol{\psi}_{\theta_2}$)
- Two target functions to estimate the SF ($\boldsymbol{\psi}_{\bar{\theta}_1}, \boldsymbol{\psi}_{\bar{\theta}_2}$)
- A policy network $\pi_\mu$
- A target policy network $\pi_{\bar{\mu}}$
- An encoder with sub-components $f_\nu, g_\nu$
- A target encoder with sub-components $f_{\bar{\nu}}, g_{\bar{\nu}}$
- A fixed target encoder with sub-components $f_{\bar{\bar{\nu}}}, g_{\bar{\bar{\nu}}}$
- A checkpoint policy $\pi_c$ and the checkpoint encoder $f_c$
- A base feature function $\phi$

**Encoder**: The encoder comprises of two sub-networks to output state and state-action embedding, such that $z^s = f_\nu(s)$ and $z^{sa} = g_\nu(z^s, a)$. The encoder is updated using the following loss:

$$\mathcal{L}_{\mathsf{Encoder}}(f_\nu, g_\nu) = \Big(g_\nu(f_\nu(s), a) - |f_\nu(s')|_\times\Big)^2 \tag{18}$$

$$= \Big(z^{sa} - |z^{s'}|_\times\Big)^2, \tag{19}$$

where $s, a, s'$ denotes the sampled transitions and $|\,.\,|_\times$ is the stop-gradient operation. Also, we represent $\bar{z}^s = f_{\bar{\nu}}(s)$, $\bar{z}^{sa} = g_{\bar{\nu}}(\bar{z}^s, a)$, $\bar{\bar{z}}^s = f_{\bar{\bar{\nu}}}(s)$, and $\bar{\bar{z}}^{sa} = g_{\bar{\bar{\nu}}}(\bar{\bar{z}}^s, a)$.

**SF network**: Motivated by standard RL algorithms (Fujimoto et al., 2018; 2023), SFM uses a pair of networks to estimate the SF. The network to estimate SF are updated with the following loss:

$$\mathcal{L}_{\mathsf{SF}}(\boldsymbol{\psi}_{\theta_i}) = \|\mathsf{target} - \boldsymbol{\psi}_{\theta_i}(\bar{z}^{sa}, \bar{z}^s, s, a)\|_2^2, \tag{20}$$

$$\mathsf{target} = \phi(s) + \frac{1}{2}\gamma * \mathrm{clip}([\boldsymbol{\psi}_{\bar{\theta}_1}(x) + \boldsymbol{\psi}_{\bar{\theta}_2}(x)], \boldsymbol{\psi}_{\min}, \boldsymbol{\psi}_{\max}), \tag{21}$$

$$x = [\bar{\bar{z}}^{s'a'}, \bar{\bar{z}}^{s'}, s', a'] \tag{22}$$

$$a' = \pi_{\bar{\mu}}(\bar{\bar{z}}^{s'}, s') + \epsilon, \text{where } \epsilon \sim \mathrm{clip}(\mathcal{N}(0, \sigma^2), -c, c). \tag{23}$$

Here, instead of taking the minimum over the SF networks for bootstrapping at the next state (Fujimoto et al., 2018), the mean over the estimates of SF is used. The action at next state $a'$ is samples similarly to TD3 (Fujimoto et al., 2018) and the same values of $(z^s, z^{sa})$ are used for each SF network. Moreover, the algorithm does clipping similar to TD7 (Fujimoto et al., 2023) on the predicted SF at the next state which is updated using $target$ (equation 32) at each time step, given by:

$$\boldsymbol{\psi}_{\min} \leftarrow \min(\boldsymbol{\psi}_{\min}, \mathsf{target}) \tag{24}$$

$$\boldsymbol{\psi}_{\max} \leftarrow \min(\boldsymbol{\psi}_{\max}, \mathsf{target}) \tag{25}$$

**Policy**: SFM uses a single policy network which takes $[z^s, s]$ as input and is updated using the following loss function described in §4.

Upon every $target\_update\_frequency$ training steps, the target networks are updated by cloning the current network parameters and remains fixed:

$$(\theta_1, \theta_2) \leftarrow (\bar{\theta}_1, \bar{\theta}_2) \tag{26}$$

$$\mu \leftarrow \bar{\mu} \tag{27}$$

$$(\nu_1, \nu_2) \leftarrow (\bar{\nu}_1, \bar{\nu}_2) \tag{28}$$

$$(\bar{\nu}_1, \bar{\nu}_2) \leftarrow (\bar{\bar{\nu}}_1, \bar{\bar{\nu}}_2) \tag{29}$$

$$\tag{30}$$

Moreover, the agent maintains a checkpointed network similar to TD7 (Refer to Appendix F of TD7 (Fujimoto et al., 2023) paper). However, TD7 utilizes the returns obtained in the environment for checkpointing. Since average returns is absent in the IRL tasks, it is not clear how to checkpoint policies. Towards this end, we propose using the negative of Mean Squared Error (MSE) between the SF of trajectories generated by agent and the SF of demonstrations as a proxy of checkpointing. To highlight some differences with the TD7 (Fujimoto et al., 2023) algorithm, SFM does not utilize a LAP (Fujimoto et al., 2020) and Huber loss to update SF network, and we leave investigating them for future research.

**Pseudo 1.** *SFM (TD7) Network Details*

**Variables:**

```
phi_dim = 128
zs_dim = 256
```

**Value SF Network:**

▷ SFM uses two SF networks each with similar architechture and forward pass.

```
l0 = Linear(state_dim + action_dim, 256)
l1 = Linear(zs_dim * 2 + 256, 256)
l2 = Linear(256, 256)
l3 = Linear(256, phi_dim)
```

**SF Network $\psi_\theta$ Forward Pass:**

```
input = concatenate([state, action])
x = AvgL1Norm(l0(inuput))
x = concatenate([zsa, zs, x])
x = ELU(l1(x))
x = ELU(l2(x))
sf = l3(x)
```

**Policy $\pi$ Network:**

```
l0 = Linear(state_dim, 256)
l1 = Linear(zs_dim + 256, 256)
l2 = Linear(256, 256)
l3 = Linear(256, action_dim)
```

**Policy $\pi$ Forward Pass:**

```
input = state
x = AvgL1Norm(l0(input))
x = concatenate([zs, x])
x = ReLU(l1(x))
x = ReLU(l2(x))
action = tanh(l3(x))
```

**State Encoder $f$ Network:**

```
l1 = Linear(state_dim, 256)
l2 = Linear(256, 256)
l3 = Linear(256, zs_dim)
```

**State Encoder $f$ Forward Pass:**

```
input = state
x = ELU(l1(input))
x = ELU(l2(x))
zs = AvgL1Norm(l3(x))
```

**State-Action Encoder $g$ Network:**

```
l1 = Linear(action_dim + zs_dim, 256)
l2 = Linear(256, 256)
l3 = Linear(256, zs_dim)
```

**State-Action Encoder $g$ Forward Pass:**

```
input = concatenate([action, zs])
x = ELU(l1(input))
x = ELU(l2(x))
zsa = l3(x)
```

## C.2 TD3-BASED NETWORK ARCHITECTURE

The architecture used in this work is inspired from the TD3 (Fujimoto et al., 2018) algorithms for continuous control tasks (Pseudo 2). We will describe the networks and sub-components used below:

- Two functions to estimate the SF ($\boldsymbol{\psi}_{\theta_1}, \boldsymbol{\psi}_{\theta_2}$)
- Two target functions to estimate the SF ($\boldsymbol{\psi}_{\bar{\theta}_1}, \boldsymbol{\psi}_{\bar{\theta}_2}$)
- A policy network $\pi_\mu$
- A base feature function $\phi$

**SF network**: Motivated by standard RL algorithms (Fujimoto et al., 2018; 2023), SFM uses a pair of networks to estimate the SF. The network to estimate SF are updated with the following loss:

$$\mathcal{L}_{\text{SF}}(\boldsymbol{\psi}_{\theta_i}) = \|\text{target} - \boldsymbol{\psi}_{\theta_i}(s, a)\|_2^2, \tag{31}$$

$$\text{target} = \phi(s) + \frac{1}{2}\gamma * (\boldsymbol{\psi}_{\bar{\theta}_1}(x) + \boldsymbol{\psi}_{\bar{\theta}_2}(x)), \tag{32}$$

$$x = [s', a'] \tag{33}$$

$$a' = \pi_\mu(s') + \epsilon, \text{where } \epsilon \sim \text{clip}(\mathcal{N}(0, \sigma^2), -c, c). \tag{34}$$

Here, instead of taking the minimum over the SF networks for bootstrapping at the next state (Fujimoto et al., 2018), the mean over the estimates of SF is used. The action at next state $a'$ is samples similarly to TD3 (Fujimoto et al., 2018). Lastly, target networks to estimate SF is updated via polyak averaging (with polyak factor of .995, given by

$$\bar{\theta}_i \leftarrow \alpha\bar{\theta}_i + (1 - \alpha)\theta_i, \quad \text{for } i = 1, 2. \tag{35}$$

**Policy**: SFM uses a single deterministic policy network which takes state $s$ to predict action $a$.

---

**Pseudo 2.** *SFM (TD3) Network Details*

**Variables:**
```
phi_dim = 128
```

**Value SF Network:**
▷ SFM uses two SF networks each with similar architechture and forward pass.
```
l0 = Linear(state_dim + action_dim, 256)
l1 = Linear(256, 256)
l2 = Linear(256, phi_dim)
```

**SF Network $\psi_\theta$ Forward Pass:**
```
input = concatenate([state, action])
x = ReLU(l0(inuput))
x = ReLU(l1(x))
x = l2(x)
```

**Policy $\pi$ Network:**
```
l0 = Linear(state_dim, 256)
l1 = Linear(256, 256)
l2 = Linear(256, action_dim)
```

**Policy $\pi$ Forward Pass:**
```
input = state
x = ReLU(l0(input))
x = ReLU(l1(x))
action = tanh(l2(x))
```

---

## C.3 SFM WITH STOCHASTIC POLICY

In light of Proposition 1, for a stochastic policy, Proposition 2 can again be used to update the policy parameters. Indeed, following the proof of Proposition 2, we have that

$$\nabla_\mu J(\pi_\mu; r^\star_{\pi_\mu \to \pi_E}) = \mathop{\mathbb{E}}_{s \sim \mathcal{B}}\left[\nabla_\mu \int_\mathcal{A} \pi_\mu(a \mid s)Q_\theta(s,a)\mathrm{d}a\right]$$

$$= \mathop{\mathbb{E}}_{s \sim \mathcal{B}}\left[\int_\mathcal{A} \boldsymbol{\psi}_\theta(s,a)^\top w^\star_{\pi_\mu \to \pi_E}\nabla_\mu \pi_\mu(a \mid s)\mathrm{d}a\right] \qquad (36)$$

$$= \sum_{i=1}^d (w^\star_{\pi_\mu \to \pi_E})_i \mathop{\mathbb{E}}_{s \sim \mathcal{B}}\left[\int_\mathcal{A} \boldsymbol{\psi}_{\theta,i}(s,a)\nabla_\mu \pi_\mu(a \mid s)\mathrm{d}a\right].$$

The integral above can be estimated without bias by Monte Carlo, using the log-derivative (RE-INFORCE) trick, or in the case of certain policy classes, the reparameterization trick (Kingma & Welling, 2014; Haarnoja et al., 2018); the latter of which tends to result in less variance in practice (Haarnoja et al., 2018), so we use it in our experiments. Towards this end, let $\mathcal{Y}$ denote a measurable space with $\varrho$ a probability measure over $\mathcal{Y}$, and suppose there exists $\mathsf{a}_\mu : \mathcal{S} \times \mathcal{Y} \to \mathcal{A}$ such that

$$\pi_\mu(\cdot \mid s) = \mathrm{Law}(\mathsf{a}_\mu(s,\epsilon)), \ \epsilon \sim \varrho.$$

Under such parameterizations, we have

$$\int_\mathcal{A} \boldsymbol{\psi}_{\theta,i}(s,a)\nabla_\mu \pi_\mu(a \mid s) = \nabla_\mu \mathbb{E}_{a \sim \pi_\mu(\cdot \mid s)}[\boldsymbol{\psi}_{\theta,i}(s,a)]$$

$$= \mathbb{E}_{\epsilon \sim \varrho}[\nabla_\mu \boldsymbol{\psi}_{\theta,i}(s, \mathsf{a}_\mu(s,\epsilon))]$$

$$= \mathbb{E}_{\epsilon \sim \varrho}[\nabla_\mu \mathsf{a}_\mu(s,\epsilon)\nabla_a \boldsymbol{\psi}_{\theta,i}(s,a)|_{a=\mathsf{a}_\mu(s,\epsilon)}].$$

**Example 1.** *Gaussian policies—that is, policies of the form $s \mapsto \mathcal{N}(m(s), \Sigma(s))$ for $m : \mathcal{S} \to \mathbb{R}^d$ and $\Sigma : \mathcal{S} \to \mathbb{R}^{d \times d}$—can be reparameterized in the aforementioned manner. Taking $\mathcal{Y} = \mathcal{A} = \mathbb{R}^d$ and $\varrho = \mathcal{N}(0, I_d)$, construct the map $\mathsf{a}_\mu : \mathcal{S} \times \mathcal{Y} \to \mathcal{A}$ by*

$$\mathsf{a}_\mu(s,\epsilon) = m_\mu(s) + \Sigma_\mu(s)\epsilon.$$

*Clearly, it holds that $\mathrm{Law}(\mathsf{a}_\mu(s,\epsilon)) = \mathcal{N}(m_\mu(s), \Sigma_\mu(s))$, accommodating any Gaussian policy.*

Altogether, our gradient estimate is computed as follows,

$$\nabla_\mu J(\pi_\mu; r^\star_{\pi_\mu \to \pi_E}) \approx \sum_{i=1}^d \hat{w}_i \frac{1}{N_1}\sum_{j=1}^{N_1} \nabla_\mu \mathsf{a}_\mu(s_{1,j}, \epsilon_{1,j})\nabla_a \boldsymbol{\psi}_{\theta,i}(s_{1,j},a)|_{a=\mathsf{a}_\mu(s_{1,j},\epsilon_{1,j})}$$

$$\hat{w} = (1-\gamma)^{-1}\frac{1}{N_2}\sum_{j=1}^{N_2}\left[\boldsymbol{\psi}_\theta(s_{2,j}, \mathsf{a}_\mu(s_{2,j}, \epsilon_{2,j})) - \gamma\boldsymbol{\psi}_{\bar\theta}(s'_{2,j}, \mathsf{a}_\mu(s'_{2,j}, \epsilon'_{2,j}))\right] - \widehat{\boldsymbol{\psi}}^E$$

$$\{s_{1,j}\}_{j=1}^{N_1} \overset{\text{iid}}{\sim} \mathcal{B}, \quad \{\epsilon_{1,j}\}_{j=1}^{N_1} \overset{\text{iid}}{\sim} \mathcal{N}(0, I_n)$$

$$\{(s_{2,j}, s'_{2,j})\}_{j=1}^{N_2} \overset{\text{iid}}{\sim} \mathcal{B}, \quad \{\epsilon_{2,j}\}_{j=1}^{N_2} \overset{\text{iid}}{\sim} \mathcal{N}(0, I_n), \ \{\epsilon'_{2,j}\}_{j=1}^{N_2} \overset{\text{iid}}{\sim} \mathcal{N}(0, I_n).$$

$$(37)$$

We note that, to compute an unbiased gradient from samples, minibatches of $(s, s')$ pairs must be sampled independently for the estimator $\hat{w}$ of $w^\star_{\pi_\mu \to \pi_E}$, and likewise action samples must be drawn independently in the computation of $\hat{w}$.

Finally, to prevent the policy from quickly collapsing to a nearly-deterministic one, we also include a policy entropy bonus in our actor updates. The architecture used in this work is inspired from the SAC (Haarnoja et al., 2018) algorithms for continuous control tasks (Pseudo 3). We will describe the networks and sub-components used below:

- Two functions to estimate the SF $(\boldsymbol{\psi}_{\theta_1}, \boldsymbol{\psi}_{\theta_2})$
- Two target functions to estimate the SF $(\boldsymbol{\psi}_{\bar\theta_1}, \boldsymbol{\psi}_{\bar\theta_2})$
- A policy network $\pi_\mu$

- A base feature function $\phi$

**SF network**: Motivated by standard RL algorithms (Fujimoto et al., 2018; 2023), SFM uses a pair of networks to estimate the SF. The network to estimate SF are updated with the following loss:

$$\mathcal{L}_{\mathsf{SF}}(\boldsymbol{\psi}_{\theta_i}) = \|\mathsf{target} - \boldsymbol{\psi}_{\theta_i}(s, a)\|_2^2,$$

$$\mathsf{target} = \phi(s) + \frac{1}{2}\gamma * (\boldsymbol{\psi}_{\bar{\theta}_1}(x) + \boldsymbol{\psi}_{\bar{\theta}_2}(x)),$$

$$x = [s', a']$$

$$a' \sim \pi_\mu(s')$$

Here, instead of taking the minimum over the SF networks for bootstrapping at the next state (Fujimoto et al., 2018), the mean over the estimates of SF is used. The action at next state $a'$ is samples similarly to TD3 (Fujimoto et al., 2018). Lastly, target networks to estimate SF is updated via polyak averaging (with polyak factor of .995, given by

$$\bar{\theta}_i \leftarrow \alpha\bar{\theta}_i + (1 - \alpha)\theta_i, \quad \text{for } i = 1, 2. \tag{38}$$

**Policy**: SFM uses a single Gaussian policy network which takes state $s$ to sample an action $a$ as described earlier.

---

**Pseudo 3.** *SFM (Stochastic) Network Details*

**Variables:**
```
phi_dim = 128
```

**Value SF Network:**
▷ SFM uses two SF networks each with similar architechture and forward pass.
```
l0 = Linear(state_dim + action_dim, 256)
l1 = Linear(256, 256)
l2 = Linear(256, phi_dim)
```

**SF Network $\psi_\theta$ Forward Pass:**
```
input = concatenate([state, action])
x = ReLU(l0(inuput))
x = ReLU(l1(x))
x = l2(x)
```

**Policy $\pi$ Network:**
```
l0 = Linear(state_dim, 256)
l1 = Linear(256, 256)
lm = Linear(256, action_dim)
ls = Linear(256, action_dim)
```

**Policy $\pi$ Forward Pass:**
```
input = state
x = ReLU(l0(input))
x = ReLU(l1(x))
mean = tanh(lm(x))
std = softplus(ls(x))
```

---

## C.4 BASE FEATURES

Since we use a base feature function $\phi$, we have two networks- 1) To provide the embedding for the state, and 2) To predict the next state from the current state and action. Pseudo 4 provides the description of the network architectures and the corresponding forward passes.

---

**Pseudo 4.** *Base Feature Network Details*

**Variables:**

```
phi_dim = 128
```

---

**Base Feature Network $\phi$ to encode state:**

```
l0 = Linear(state_dim, 512)
l2 = Linear(512, 512)
l3 = Linear(512, phi_dim)
```

**Base Feature $\phi$ Forward Pass:**

```
input = state
x = Layernorm(l1(x))
x = tanh(x)
x = ReLU(x)
phi_s = L2Norm(l3(x))
```

---

**FDM Network:**

```
l0 = Linear(phi_dim + action_dim, 512)
l1 = Linear(512, 512)
l2 = Linear(512, action_dim)
```

**FDM Network Forward Pass:**

```
input = concatenate([phi_s, action])
x = ReLU(l0(x))
x = ReLU(l1(x))
action = tanh(l2(x))
```

---

## C.5 STATE-ONLY ADVERSARIAL BASELINES

For the state-only MM method, we used the same architecture as TD7 (Fujimoto et al., 2023) or TD3 (Fujimoto et al., 2018) for the RL subroutine. We kept the same architecture of the discriminator as provided in the official implementation. However, we modified the discriminator to take only states as inputs. Additionally, we used gradient penalty and learning rate decay to update the discriminator, and OAdam optimizer (Daskalakis et al., 2018) for all networks. For GAIfO (Torabi et al., 2018), we used the same architecture as state-only MM. However, the discriminator takes the state-transition denoted as the state and next-state pair as input.

## D HYPERPARAMETERS

In Table 5, we provide the details of the hyperparameters used for learning. Many of our hyperparamters are similar to the TD7 (Fujimoto et al., 2023) algorithm. Important hyperparameters include the discount factor $\gamma$ for the SF network and tuned it with values $\gamma = [0.98, 0.99, 0.995]$ and report the ones that worked best in the table. Rest, our method was robust to hyperparameters like learning rate and batch-size used during training.

| Name | Value |
|---|---|
| Batch Size | 1024 |
| Discount factor $\gamma$ for SF | .99 |
| Actor Learning Rate | 5e-4 |
| SF network Learning Rate | 5e-4 |
| Base feature function learning Rate | 5e-4 |
| Network update interval | 250 |
| Target noise | .2 |
| Target Noise Clip | .5 |
| Action noise | .1 |
| Environments steps | 1e6 |

Table 5: Hyper parameters used to train SFM.

