# OpenReview forum: "Non-Adversarial Inverse Reinforcement Learning via Successor Feature Matching"
_ICLR.cc/2025/Conference — ICLR 2025 Poster_

### Official Review · Reviewer_ZRQG · 2024-10-24

**Soundness:** 3
**Presentation:** 2
**Contribution:** 2
**Rating:** 3
**Confidence:** 3

**Summary:**

This paper proposes Successor Feature Matching (SFM) in the Inverse Reinforcement Learning (IRL) problem.  The authors focused generalized reward-like properties of  successor features (Barreto et al. 2017), e.g. recovering reward, discounted accumulation. These allowed SFM for alternative formulation of policy optimization using L2 loss in Eq. (4). The experimental results suggest that the proposed SFM yields good scores in single-demonstration benchmarks and much more impressive scores among state-only IRL algorithms.

**Strengths:**

1. The proposed SFM, in contrast to adversarial methods, is more rooted in the historical context of early IRL studies.
2. In my opinion, IRL (or even RL) methods do not have to restrict themselves to formulating scalar reward signals. Adaptation SF (Barreto et al., 2017) expands the notion of IRL into feature learning, which can relax some rigidity in RL-IRL formulation. Since the features have successfully formulated vectorized signals in the transfer learning tasks in RL, this could be a significant contribution that relaxes studies of IRL.
3. The proposed method achieved good performance among modern IRL methods.
4. Propositions 1-3 are straightforward to understand.

**Weaknesses:**

1. I do not understand why "non-adversarial" property is framed as the main contribution throughout the paper. Adversarial learning and SFM are not mutually exclusive. Eq. (3) shows that the base feature can be trained with adversarial learning by setting $\mathcal{L}_\mathrm{feat} = \mathrm{Eq. (3)}$. If the author intended to deepen our understanding of the role of adversarial learning in IRL, they should have provided (1) another theoretical justification of non-adversarial learning and (2) ablation studies, including Eq. (3) in Fig. 6, and (3) results of combining various base feature losses that verifies adversarial losses might deteriorate IRL performance.
2. Compared to Figs. 4 and 5, single-demonstration task performance in Fig 3 is only sometimes good, suggesting that SF is more focused on representing states. For some scenarios, action representation performance might be necessary.
3. To demonstrate scalability, including more complex (or demonstrative) benchmarks in Fig. 3 would be beneficial to IRL researchers to grasp the model's performance.
4. There are no supporting tables in the appendix that measure the performance of the learning curves in Figs. 3, 5, 6.

**Questions:**

1. How applying l2-loss  in Eq.(4) can be understood for the policy $\pi_\mu$?
2. I think SFM (random) performed well; what is the reasoning behind this and presenting Eqs? (9-11)? Could the base feature only trained with arbitrary unsupervised learning with only states without the next states, actions, or goals?

---

> ### Author Response · Authors · 2024-11-18
> **Rebuttal**
>
> We thank the reviewer for their careful review and their thought-provoking questions. We respond to the questions below:
>
> **W1: "non-adversarial” property and base feature trained with adversarial learning by setting $\mathcal{L}_{feat}$ = Eq.(3)**\
> In Eq 3, we show how the task of IRL which uses adversarial learning can be framed as minimizing the $L_2$ gap between expected features, leading to a framework that can use any mechanism to learn base features.
>
> Regarding the non-adversarial property-- you are of course correct that our framework does not prohibit $\mathcal{L}_{feat}$ from being trained adversarially. However, our argument is a sort of converse-- the framework enables us to *avoid* adversarial feature learning, which leads to simpler algorithms, achieving greater performance than the adversarial baselines.
> That said, your hypothesis that adversarially-trained features can lead to even *better* performance is interesting.
>
> Suppose one were to employ an adversarially-trained $\mathcal{L}_{feat}$-- as you suggest, one can do this via an adversarial IRL algorithm similar to MM. There are two possible outcomes that we might observe by training SFM with those features:
> 1. The performance is worse than our non-adversarial SFM results. In this case, we conclude that non-adversarial features should be preferred.
> 2. The performance is better than what we achieved in SFM. In this case, since SFM already generally outperforms baselines, we can use adversarial features to get better IRL agents, which again is a positive outcome.
>
> So, in any case, we argue that SFM is a valuable tool for IRL.
> Continuing on this thread, we conduct experiments with adversarially (Adv) trained features.
> We observe in Fig 5 of the updated paper that adversarial features do well, but FDM still outperforms other base feature functions.
> (Note that we moved Fig 5 in the submitted paper to the Appendix due to space constraints, Fig 6 is now Fig 7 and Fig 5 presents the RLiable plot of base feature functions in the revision).
>
> **W2: Compared to Figs. 4 and 5, single-demonstration task performance in Fig 3 is only sometimes good**\
> We have provided agents with single expert demonstrations in all our experiments.
> The major change between Fig 3 and Fig 5 is the architecture of the policy optimizer.
> In Figure 3, we compared the performance of SFM with state-only MM and GAIfO with a framework based on TD7 algorithm.
> Our implementation of baselines with TD7 framework significantly improved the performance when compared to using a TD3 optimizer (Fig 4 & 5) which explains the large gaps in performance for the adversarial baselines between Fig 3 and Fig 5.
> However, the performance of SFM is similar with both frameworks suggesting that SFM is more robust to policy optimizers.
>
> In this work, we focus on the scenario of state-only demonstrations and use state-only base features.
> Without expert actions, it is challenging to estimate the state-action base features required to estimate the expert's SF.
> However, we would like to highlight that it is not a limitation and SFM can be easily extended to have state-action features when using demonstrations with expert action labels (as discussed between lines 282-288 in the paper).
>
> **W4: no supporting tables in the appendix**\
> We have added tables in Appendix D showing the numerical results of the Figures.
>
> **Q1: How applying l2-loss in Eq.(4) can be understood for the policy $\pi_{\mu}$?**\
> For the policy $\pi_{\mu}$, the l2-loss defined in objective will optimize the agent to match the expected features with the expert.
> As we observe in Eq 3, this would provide bounds of the performance gap between the expert and the agent, i.e. if the l2 gap is small, then the imitation gap between the expert and the learning agent should be small.
> To compute the gradients of the policy for this l2-based objective, we propose leveraging Proposition 1 \& 2 that provides the gradients for any states sampled from the replay buffer.

---

> > ### Author Response · Authors · 2024-11-18
> > **Rebuttal**
> >
> > **Q2: Reasoning behind random feature working well?**\
> > In many RL tasks [1,2], random features have shown promising results for learning agents.
> > This was observed in our experiments too where SFM (random) is generally performant, but we hypothesized that leveraging environmental structure in the base features could lead to improved performance; this is why we present a general framework that enables feature learning. Our hypothesis was correct, as we showed that using FDM features tends to produce superior results.
> >
> > **Q2: base feature with arbitrary unsupervised learning with only states without the next states, actions, or goals?**\
> > Yes, the base features can be trained with arbitrary unsupervised learning with only states.
> > As an experiment, we trained an Autoencoder (AE) over the states and used the latent embedding as the base features (Fig 5 of updated paper and Eq 9 describe the approach).
> > Although we compared with a few different ways to learn base features, our method can be extended to use any feature-learning method or even utilize pre-trained features.
> >
> > We would be happy to answer any further questions.
> > We hope we addressed most of your concerns and you consider increasing the score.
> >
> > References
> >
> > [1] Seo et al., State Entropy Maximization with Random Encoders
> > for Efficient Exploration, ICML 2021.
> >
> > [2] Farebrother et al., Proto-Value Networks: Scaling Representation Learning with Auxiliary Tasks, ICLR 2023.

---

> ### Author Response · Authors · 2024-11-25
>
> Dear Reviewer ZRQG,
>
> Thank you for your time and feedback, which helped improve the paper. During the rebuttal period, we added comparisons with other base feature functions (Fig 5): 1) Autoencoder over only states and not next states, actions, or goals (Q2); and 2) Adversarial where the base features are trained with Eq 3 (W1).
>
> We believe our revision and responses have addressed most of your concerns. As the author-reviewer discussion period ends in 2 days, we would be happy to discuss and address any further questions during this time.
>
> Best Regards,\
> Authors

---

> ### Comment · Reviewer_ZRQG · 2024-11-25
> **Thanks for the rebuttal**
>
> Deal authors,
>
> I appreciate your detailed explanation and additional experiments. In particular, I am impressed that the authors took W1 and Q2 into serious consideration in the revision. Currently, I have follow-up questions.
> * In revised Fig 5, it seems the new Adv version performs in second place. Does this fit well with your original claims, considering the adv version might run slightly more efficiently without acquiring next-states? I still have a minor concern that the "Non-adversarial IRL" title might be misleading.
> * I can agree that this method might be suitable for certain RL envs with physics simulations. What are your current thoughts on whether SFM could work well for other environments, such as high-dim vision-based and real-world linguistic domains?

---

> ### Author Response · Authors · 2024-11-25
>
> Dear Reviewer ZRQG,
>
> We thank you for the response and address the questions below:
>
> **Does this fit well with your original claims**\
> While adversarial features performed reasonably well, we would like to emphasize that this does not go against our claims. In particular, non-adversarial features achieve equally good (and often better) performance, and as we discuss next, they do so without sacrificing efficiency.
>
> **Adv version might run slightly more efficiently without acquiring next-states?**\
> Firstly, training adversarial features is quite challenging in general. For our experiments, we had to tune multiple parameters like the gradient penalty coefficient, learning rate decay, and updating the features quite less frequently as compared to the policy.
> Secondly, for training the adversarial features, the agent collects on-policy samples by generating trajectories with the current policy (inspired from adversarial reward learning approaches like GAIL [1] and HyPE [2]).
> The states visited in this on-policy dataset are used for updating the adversarial features (as described in equation 13).
> However, for training non-adversarial features, the agent could just learn it from transitions in the replay buffer without additional samples which makes it more efficient at learning.
> We tried learning adversarial features by sampling states from the replay buffer, but it was not performing well in our experiments.
> Therefore, non-adversarial feature learning is more efficient.
>
> **Whether SFM could work well for other environments?**\
> This is of course an interesting question. Classically, IRL methods have struggled on such domains, especially in the state-only setting. However, by reducing the IRL problem to RL, we hope that we can achieve strong performance in such domains by leveraging recent representation learning techniques in RL. Given the promising results of RL in such domains, we believe this is an intriguing avenue for future work.
>
> We would be happy to answer any further questions and hope you consider increasing the score.
>
> Best Regards,\
> Authors
>
> ### References
> [1] Ho et al., Generative Adversarial Imitation Learning, NeurIPS 2016.\
> [2] Ren et al., Hybrid Inverse Reinforcement Learning, ICML 2024.

---

> > ### Author Response · Authors · 2024-11-29
> >
> > Dear Reviewer ZRQG,
> >
> > We thank you for the responses and look forward to discussions with you. Continuing on the previous response:
> >
> > - In this work, we show that imitation policies can be learned by minimizing the gap between successor features of expert and agent. Interestingly, our method is agnostic to the choice of base feature function and can use both adversarial or non-adversarial approaches for learning (as presented during the revision). Thereby, SFM can also utilize pre-trained features which we leave for future work.
> > - By leveraging recent techniques in unsupervised RL on base feature functions to estimate SFs, we show SFM can be trained without adversarial training (given that FDM performs best across base-feature function). However, prior methods like MM, GAIfO cannot get around this bi-level optimization-- as they heavily rely on a reward function trained adversarially for updating the agent’s policy.
> > - Since SFM works best with non-adversarial features, we emphasize our approach as a method for *IRL with state-only demonstrations that can learn without bi-level optimization*.
> > Our experiments show that SFM works well compared to prior methods on tasks from DMControl Suite-- validating that reduction of IRL to RL works well and as highlighted in your strengths can relax some rigidity in RL-IRL formulation.
> >
> > We hope this answers to your concerns and we are available to address any questions or concerns you may have.
> >
> > Best Regards,\
> > The Authors

---

> > > ### Comment · Reviewer_ZRQG · 2024-12-03
> > >
> > > Dear authors,
> > >
> > > Thank you for your detailed response. Based on your response and the reviews from other reviewers, I might need some more time to make my final recommendation. I would like to finalize my score after the discussion with the other reviewers.

---

> > > > ### Author Response · Authors · 2024-12-03
> > > >
> > > > Dear Reviewer ZRQG,
> > > >
> > > > Thank you once again for your time and valuable suggestions. We truly appreciate the engaging analysis they inspired. As the discussion period draws to a close, we hope our responses and proposed changes have resonated with you.
> > > >
> > > > Best Regards,\
> > > > The Authors

---

### Official Review · Reviewer_YndG · 2024-10-30

**Soundness:** 3
**Presentation:** 3
**Contribution:** 4
**Rating:** 8
**Confidence:** 2

**Summary:**

The paper introduces Successor Feature Matching (SFM), a novel method to Inverse Reinforcement Learning (IRL) which is non-adversarial and does not require expert action labels. SFM directly optimizes a policy to match the expert's successor features (SFs) using policy gradient ascent. The paper claims that SFM offers several advantages over existing IRL methods, including simplified training, state-only learning and robustness to optimizer choice.
The paper supports these claims through empirical evaluations on the DeepMind Control suite, demonstrating that SFM outperforms state-of-the-art adversarial and non-adversarial baselines on a variety of single-demonstration tasks. Additionally, the paper explores the impact of different base feature functions on SFM's performance, finding that Forward Dynamics Models (FDM) lands the best results.

**Strengths:**

- **Novel and well-motivated approach:** SFM provides a new angle in IRL by leveraging SFs for direct policy optimization, offering an alternative to adversarial methods.
- **Strong empirical results:** Experiments demonstrate the superiority of SFM over sota methods on normalized return and optimality gap across single-demonstration tasks, demonstrating its effectiveness.
- **State-only learning:** The ability to learn from state-only demonstrations is a great contribution.
- **Robustness to RL optimizer choice:** SFM's performance with both strong (TD7) and weaker (TD3) optimizers shows versatility, making it potentially useful for resource-limited applications.
- The experiments are conducted in clear logic and the analysis and observations are novel and interesting.

**Weaknesses:**

- **Dependence on deterministic policy gradients:** The current formulation of SFM is tied to deterministic policy gradient algorithms. Exploring extensions to stochastic policies would broaden its scope. However, it's clearly mentioned by the authors in the discussion part.
- **Exploration Challenges:** as stated by the authors, the presented method does not fully resolve IRL exploration issues. I wonder how the performance may vary significantly across domains requiring extensive exploration.

**Questions:**

- How does SFM handle scenarios where expert demonstrations do not cover the entire state space, particularly in sparse-reward or high-dimensional tasks?
- Can the authors elaborate on potential extensions of SFM to stochastic policy optimizers, and any early experiments indicating feasibility?

---

> ### Author Response · Authors · 2024-11-18
> **Rebuttal**
>
> We thank the reviewer for their thoughtful review and their enthusiasm in our work. With regard to both of your questions, these are valid points, and we have elaborated on extending SFM to stochastic policies as well as exploration in our general response.
>
> We hope we addressed most of your concerns and would be happy to answer any further questions.

---

> > ### Comment · Reviewer_YndG · 2024-11-26
> >
> > Thank you for your responses. After reading the discussions with the other reviewers, I will maintain my positive evaluation.

---

### Official Review · Reviewer_w2Zs · 2024-11-02

**Soundness:** 3
**Presentation:** 3
**Contribution:** 3
**Rating:** 6
**Confidence:** 3

**Summary:**

This paper introduces Successor Feature Matching (SFM), a novel non-adversarial approach to Inverse Reinforcement Learning (IRL). The key innovation is reformulating IRL as a direct policy optimization problem that matches successor features between the agent and expert demonstrations. The method has three notable contributions:

1. A non-adversarial approach that avoids the computational expense and instability of traditional adversarial IRL methods
2. The ability to learn from state-only demonstrations without requiring expert action labels
3. Strong performance from as little as a single expert demonstration

The method leverages successor features to estimate expected cumulative features and uses policy gradient descent to minimize the gap between learner and expert features. The authors demonstrate superior performance compared to both adversarial and non-adversarial baselines across multiple control tasks.

**Strengths:**

1. The non-adversarial approach using successor features is novel and elegantly simple compared to existing methods
2. The approach is well-justified with clear theoretical analysis and proofs
3. The method works with state-only demonstrations and single examples, making it widely applicable
4. SFM integrates seamlessly with existing actor-critic frameworks
5. Demonstrates superior performance across multiple tasks and metrics
6. Shows consistent performance even with weaker policy optimizers

**Weaknesses:**

1. The paper doesn't fully address how SFM handles the exploration problem inherent in IRL
2. The current implementation is tied to deterministic policy gradients, potentially limiting its applicability
3. The method currently only works with state-only base feature functions
4. While comprehensive, the evaluation is limited to DMC suite tasks; testing on more diverse environments would strengthen the claims

**Questions:**

1. How would the method perform on tasks with sparse rewards or requiring significant exploration?
2. Could the approach be extended to work with stochastic policies?
3. How does the method compare to recent offline IRL approaches?
4. What are the computational requirements compared to adversarial methods?
5. How sensitive is the performance to the choice of successor feature architecture?

---

> ### Author Response · Authors · 2024-11-18
> **Rebuttal**
>
> We thank the reviewer for their thorough assessment of our work and their insightful comments.
> We have provided an extension to stochastic policies and discussed about exploration in the general response.
> We respond to the remaining questions below:
>
> **W3: State only base features**\
> In this work, we train SFM with state-only base features to deal with state-only demonstrations; if we have action labels in expert demonstrations, SFM can easily support action-dependent base features.
> Indeed, the only area where state-only base features are required is where we estimate the expert's SFs: if we have expert actions, we can estimate SFs for action-dependent base features.
> This is a fairly natural constraint---without access to expert actions, it is unlikely that learned action-conditioned base features will lead to improved imitation.
>
> **Q3: Comparison with offline IRL approaches**:\
> While SFM is an online IRL algorithm, it makes it hard to compare with offline methods.
> Typically, interactive methods are proven theoretically to have better performance as they deal with covariate shift by allowing agents to recover from its own mistakes.
> Moreover, offline methods assume presence of optimal or sub-optimal dataset whereas this work provides agents access only to a single offline trajectory.
> In this regard, we skipped comparison with offline IRL methods in this work.
> We agree an interesting direction of future work is an offline version of SFM by borrowing tricks from the offline RL literature.
> Lastly, we compared with Behavior Cloning (BC) as an offline imitation learning baseline where the agent was not able to learn with a single trajectory with the expert action information in demonstrations.
>
> **Q4: Computational requirements vs adversarial methods**:\
> In terms of computational complexity, SFM and adversarial methods are fairly similar---per training iteration, SFM updates its SFs and its policy, while adversarial methods would update a discriminator/reward function and its policy. However, since the SF and policy objectives in SFM do not form a min-max game, the two gradient updates can occur simultaneously---we do not need to consider complicated training dynamics like in adversarial methods, which induce more hyperparameters that can be tricky to tune.
>
> **Q5: Sensitivity to SF architecture**:\
>  In all of our experiments, we simply used the standard architecture used for training the TD3 critic (with a modified head to predict multivariate features). We did not rigorously evaluate the sensitivity in this regard, however our experiments show that the most natural architecture choice tends to work very well.
>
> We would be happy to answer any further questions.
> We hope we addressed most of your concerns and you consider increasing the score.

---

> ### Author Response · Authors · 2024-11-25
>
> Dear Reviewer w2Zs,
>
> Thank you for your time and feedback, which helped improve the paper. During the rebuttal period, we introduced a variant of SFM incorporating stochastic policies (addressing Q2 and W2) and expanded our discussion on extending SFM to address exploration challenges in reinforcement learning (as part of our response to Q1 and W1).
>
> We believe our revision and responses have addressed most of your concerns. As the author-reviewer discussion period ends soon, we would be happy to address any further questions during this time and hope you consider increasing the score.
>
> Best Regards,\
> Authors

---

> > ### Author Response · Authors · 2024-12-02
> >
> > Dear Reviewer w2Zs,
> >
> > We are available to answer any questions you may have and look forward to continuing our discussion with you.
> >
> > Best Regards,\
> > The Authors

---

### Official Review · Reviewer_7726 · 2024-11-04

**Soundness:** 2
**Presentation:** 3
**Contribution:** 3
**Rating:** 6
**Confidence:** 3

**Summary:**

This work considers inverse RL in the state-only setting. Leveraging the linear structure of returns, as inner product of successor features and reward weight, the authors could model inverse RL without adversarially optimizing the reward weight, while is often done in prior works. This allows one to directly optimize policy through minimizing the gap between the successor features of expert policy and of the learned policy.

**Strengths:**

- The paper is well organized and easy to follow

- The motivation is sound and the resulting algorithm is straightforward (without requiring min-max optimization)

**Weaknesses:**

- It is quite surprising that IQL failed most of the tasks in Figure 3

- The choice of baselines could be expanded. While SFM considers the state-only setting, GAIfO is the only baseline that originally proposed for this setting. It might be reasonable to include a few baselines that are designed for state-only setting, for example [1, 2] (with public implementation if my memory serves me right) and some more up-to-date baselines are appreciated.

----

Note: I have not actively followed the recent literature on successor features, so I am uncertain about its novelty when applied to the IRL setting. As a result, I have a low confidence score.

[1] Gangwani, Tanmay, and Jian Peng. "State-only imitation with transition dynamics mismatch." arXiv preprint arXiv:2002.11879 (2020).
[2] Zhu, Zhuangdi, et al. "Off-policy imitation learning from observations." Advances in neural information processing systems 33 (2020): 12402-12413.

**Questions:**

- what are the hyperparameter search space for SFM and baselines respectively?

---

> ### Author Response · Authors · 2024-11-21
> **Rebuttal**
>
> We thank the reviewer for their thoughtul review and their enthusiasm about our work. We respond to the questions below:
>
> **W1: Performance of IQ-Learn**\
> As discussed in Jena et al. [3], some imitation learning methods can hack the rewards using the terminal states and survival instincts.
> However, we conduct experiments on DMControl suite where each task is infinite horizon tasks with fixed episode length-- a scenario where agents cannot hack through survival biases.
> We believe this might be true for IQ-Learn due to which the method does not perform well on infinite horizon tasks.
> Note that recent works [4] have demonstrated similar performance of IQ-Learn with the DMControl tasks.
>
> **W2: The choice of baselines could be expanded**\
> We appreciate your suggestions about further baselines to include.
> We do note, that we actually significantly strengthened the baselines in our work by incorporating the latest techniques in deep RL / IRL, massively improving performance over their original versions.
> We have added OPOLO [2] as another baseline in our revision, and while it performed competitively with other baselines, it struggled on the quadruped tasks---ultimately, SFM still outperforms all baselines.
> Our hypothesis is that OPOLO heavily depends on a good Inverse Dynamics Module (IDM) which is hard to learn in the quadruped domain (something we saw in our experiments in Fig 7 of updated draft).
> I2L [1] is an interesting method that deals with expert demonstrations collected in a different MDP than the learner's MDP to tackle challenges with changing dynamics.
> However, this is not the setting we tackle in our paper; indeed, none of the baseline methods are designed to perform well in this setting, and it is not the setting of our experiments.
> That said, we believe similar tricks from I2L can be leveraged to learn base features that are agnostic to transition dynamics to extend SFM to this setting, and for now we leave it as a very compelling direction for future work.
>
> **Q: What are the hyperparameter search space for SFM and baselines respectively?**\
> For adversarial baselines, we observed that the choice of policy optimizer (between TD7 and TD3) significantly affected the performance.
> Other hyperparameters for tuning the discriminator (reward model) include the coefficient of gradient penalty loss, learning rate decay and frequency of updates during training.
> For SFM, the SF network uses the standard architecture used for training the TD3 critic (with a modified head to predict multivariate features).
> For base feature functions, we directly use the architecture and implementation provided in Hilp~[5].
> The hyperparameters for SFM include the policy optimizer, base feature function and the dimension of base features.
> We found that SFM works ``out of the box'' with the most natural choice of hyparparameters (namely, architecture of the base feature function and policy optimizer).
> As such, we did not extensively assess the sensitivity to these parameters, because we found empirically that they do not need to be tuned.
> Finally, we stress that SFM is fairly robust to hyperparameters and policy optimizers across environments.
>
> We believe our revision has addressed all of your comments, and if so, we would be grateful if you would consider increasing your score.
> Otherwise, we are eager to discuss if you have any further questions or comments.
>
> ### References
> [1] Gangwani et al., State-only imitation with transition dynamics mismatch, ICLR 2020.\
> [2] Zhu et al., 'Off-policy imitation learning from observations.', NeurIPS 2020.\
> [3] Jena et al., Addressing reward bias in adversarial imitation learning with neutral reward functions, arXiv:2009.09467.\
> [4] Pirotta et al., Fast Imitation via Behavior Foundation Models, ICLR 2023.\
> [5] Park et al., Foundation Policies with Hilbert Representations, ICML 2024

---

> > ### Comment · Reviewer_7726 · 2024-11-25
> >
> > Thank you for the responses. I'll keep my (positive) evaluation.

---

### Author Response · Authors · 2024-11-18
**Rebuttal: General Response**

We thank all reviewers for their insightful comments and their constructive remarks. In this general comment, we would like to address two questions brought up by most of the reviewers, namely 1) how can exploration be addressed in the SFM framework, and 2) whether SFM can accommodate stochastic policies.

**Stochastic Policies**\
We introduced SFM in the context of deterministic policies mainly for the sake of simplicity. However, SFM can be extended to accommodate stochastic policies (and stochastic policy optimizers) without much difficulty. We implemented a SFM variant with a stochastic policy parameterized with a Gaussian distribution. In Appendix E of updated draft, we added the variant with a pseudocode in Algorithm 2.
Our experiments show that SFM can also learn to imitate with stochastic policies and achieve comparable performance with deterministic policy optimizers.
Notably, however, in the context of IRL from a single expert demonstration, the motivation for learning a stochastic policy is relatively unclear. The only benefit to supporting stochastic policies that we can think of is that it accommodates a broader class of policy optimization subroutines, which we agree is useful.

**Exploration**\
We believe SFM can leverage any exploration strategy common in both IRL and RL to speed up learning. For IRL, recent works either reset to the expert visited states [1] or leverage a hybrid learning approach [2].
The former method requires a simulator where the agent can be resetted to any given state making in infeasible for many scenarios.
Whereas, hybrid approaches leverage a hybrid replay buffer which have shown positive results in both IRL [2] and RL tasks [3].
A challenge with hybrid approaches is that they assume the action information in the demonstrations when sampling an expert transition.
It is an interesting direction to leverage such strategies for state-only setting, however we leave it for future work.

Currently, SFM incorporates exploration much like TD3 by adding noise to actions during training.
Of course, this type of exploration strategy does not generally incur low regret, but it is not any less principled than actor-critic methods applied to deep RL.
We highlight that SFM can leverage any existing exploration strategies common for RL tasks.
A potential direction is to use the successor features learned by SFM to derive exploration bonuses--- with several works demonstrating strong results using successor features for this purpose [4].

To summarize, we agree exploration is an important concern when scaling to harder tasks.
However, this work focuses on coming up with an RL-like training of agents for imitation learning and we believe in doing so, our method can leverage any strategic exploration algorithm.

### References:
[1] Ren et al., Hybrid Inverse Reinforcement Learning, ICML 2024.\
[2] Swamy et al., Inverse Reinforcement Learning without Reinforcement Learning, ICML 2023.\
[3] Ball et al., Efficient Online Reinforcement Learning with Offline Data, ICML 2023.\
[4] Machado et al., Count-Based Exploration with the Successor Representation, AAAI 2020.

---

### Meta-Review · Area_Chair_PNSK · 2024-12-19

**Metareview:**

This paper studies Inverse Reinforcement Learning (IRL) via a successor-feature-based approach, the Successor Feature Matching (SFM),  which is non-adversarial and does not require expert action labels. In particular, SFM directly optimizes a policy to match the expert's successor features (SFs) using policy optimization. The effectiveness of the proposed approach was then supported by experiments, which showed that SFM outperformed state-of-the-art adversarial and non-adversarial baselines, with low computational cost and sample complexity (even from single-demonstration trajectories), and state-only demonstration data. The technical idea is novel, the problem studied is very well-motivated, and the paper is overall well-written. The main concerns were regarding the use of terminology, the applicability to general stochastic policies, and how to incorporate exploration. I believe the merits outweigh the drawbacks of the paper, and suggest the authors incorporate the feedback from this round in preparing the camera-ready version of the paper.

**Additional Comments On Reviewer Discussion:**

There were some concerns regarding the misuse of the terminology "non-adversarial", the limitation to studying only deterministic policies, and the lack of efficient "exploration" in the method, together with some clarity/presentation issues for the experimental results. The authors acknowledged the comments, and addressed most of them by supplementing new experiments on Gaussian stochastic policies in the updated paper, discussing how to incorporate exploration, and explaining the use of "non-adversarial". Overall I think the responses were reasonable and satisfying, and the terminology issue is not highly critical.

---

### Decision · Program_Chairs · 2025-01-22

Accept (Poster)